# CURI: A BENCHMARK FOR PRODUCTIVE CONCEPT LEARNING UNDER UNCERTAINTY

## ABSTRACT

Humans can learn and reason under substantial uncertainty in a space of infinitely many concepts, including structured relational concepts ("a scene with objects that have the same color") and ad-hoc categories defined through goals ("objects that could fall on one's head"). In contrast, standard classification benchmarks: 1) consider only a fixed set of category labels, 2) do not evaluate compositional concept learning and 3) do not explicitly capture a notion of reasoning under uncertainty. We introduce a new few-shot, meta-learning benchmark, Compositional Reasoning Under Uncertainty (CURI) to bridge this gap. CURI evaluates different aspects of productive and systematic generalization, including abstract understandings of disentangling, productive generalization, learning boolean operations, variable binding, etc. Importantly, it also defines a model-independent "compositionality gap" to evaluate difficulty of generalizing out-of-distribution along each of these axes. Extensive evaluations across a range of modeling choices spanning different modalities (image, schemas, and sounds), splits, privileged auxiliary concept information, and choices of negatives reveal substantial scope for modeling advances on the proposed task. All code and datasets will be available online.

## 1 INTRODUCTION

Human concept learning is more flexible than today's AI systems. Human conceptual knowledge is *productive*: people can understand and generate novel concepts via compositions of existing concepts ("an apartment dog") (Murphy, 2002), unlike standard machine classifiers that are limited to a fixed set of classes ("dog", "cat", etc.). Further, humans can induce goal-based, "ad hoc" categories such as "things to take from one's apartment in a fire" (children, dogs, keepsakes, etc.) (Barsalou, 1983). Thus, unlike AI systems, humans reason seamlessly in large, essentially "unbounded" concept spaces.

Beyond unboundedness, a natural challenge in such concept spaces is *uncertainty* – the right concept to be inferred is uncertain, as a plethora of candidate concepts could explain observations. For *e.g.* in Figure 1 (top, image panel), the "right" concept could be that "All objects are blue and have the same size", but it could also be "There are less than four objects in the scene", or "All objects have the same color". Humans gracefully handle such uncertainty and underdetermination (Tenenbaum & Griffiths, 2001; Xu & Tenenbaum, 2007; Goodman et al., 2008; Piantadosi et al., 2016). Popular compositional reasoning benchmarks such as CLEVR (Johnson & Zhang, 2016) for visual question answering and Ravens Progressive Matrices (Santoro et al., 2017) for deductive, analogical reasoning are compositionally rich and challenging in nature, but do not tackle ambiguity and underdetermination.

We address this gap in the literature, and propose the Compositional Reasoning Under Uncertainty (CURI) benchmark to study how modern machine learning systems can learn concepts spanning a large, productively defined space (Figure 1). In pursuit of this goal, we instantiate a meta learning task where a model must acquire a compositional concept from finite samples. A signature of productivity in human thought is our ability to handle novel combinations of known, atomic components. Thus, in CURI we instantiate different systematic train-test splits to analyze different forms of generalization in concept learning, involving novel combinations of intrinsic properties (*e.g.* color, shape) with boolean operators, counting, extrinsic object properties (*e.g.* object location), and a novel test of variable binding in context of compositional learning.

While related systematic splits have been proposed in prior work in context of other tasks such as question answering and analogical reasoning (Barrett et al., 2018; Hill et al., 2019; Agrawal et al.,

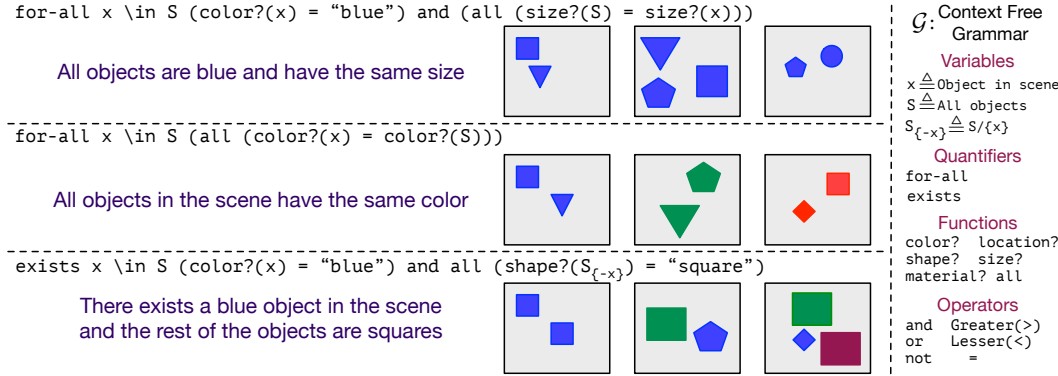

Figure 1: **Concept Space**. Three example concepts (rows) along with schematic positive examples. Actual scenes are rendered in multiple ways including the CLEVR renderer (Johnson et al., 2016) (see Figure 2). **Right:** The grammar of variables, quantifiers, functions and operators to induce compositional concepts.

2017; Johnson et al., 2016; Vedantam et al., 2017; Higgins et al., 2017; Bakhtin et al., 2019; Lake & Baroni, 2018; Ruis et al., 2020), ours is the first benchmark which tests different qualitative aspects of reasoning about productive concepts under uncertainty.

**Compositional Reasoning Under Uncertainty (CURI) Task.** Concretely, the CURI task tests few-shot learning of relational concepts in a large compositional conceptual space, with design inspiration from studies in cognitive modeling using a language of thought (LOT) approach (Fodor, 1975; Piantadosi, 2011; Kemp et al., 2005). CURI includes scene-based concepts such as "All objects have the same color" and "There exists a blue object while the rest are triangles" (Figure 1) but unlike CLEVR (Johnson et al., 2016) there are too few examples to deduce answers with certainty. Our benchmark is defined through a series of meta-learning episodes (see example in Figure 2): given positive and negative examples of a new concept $D_{\text{supp}}$ (known as the "support set"), the goal of an episode is to classify new examples $D_{\text{query}}$ (the "query set"). As in few-shot classification (Fei-Fei et al., 2006), meta-learning (Vinyals et al., 2016), and other open-set tasks (Lampert et al., 2014), models are evaluated on novel classes outside the (meta-)training set. Unlike previous work (Triantafillou et al., 2019; Lake et al., 2019) that focuses on atomic concepts, our benchmarks concerns more structured, relational concepts built compositionally from a set of atomic concepts, and involves reasoning under uncertainty – an ideal learner must marginalize over many hypotheses when making predictions (Gelman et al., 2004; Xu & Tenenbaum, 2007; Piantadosi et al., 2016).

We also vary the modality in which scenes are presented—rendering them as images, symbolic schemas, and sounds— enabling future research on modality-specific representational choices for compositional reasoning under uncertainty. Finally, we vary the concepts learned by the model during meta-training and meta-testing to test different aspects of systematic generalization.

**Compositionality Gap.** In addition to defining systematic splits, we also characterize (for the first time, in our knowledge), the difficulty of

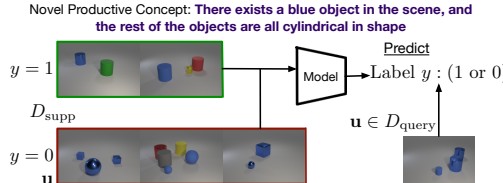

Figure 2: **CURI Task**. Given as input a support set $D_{\text{supp}}$, with positive and negative examples corresponding to concept, the model has to infer the concept and produce accurate predictions on novel images (right).

generalization entailed by each split by introducing the notion of a model-independent "compositionality gap". Concretely, the compositionality gap is the difference in test performance between an ideal Bayesian learner with access to the full hypothesis space, and a Bayesian learner with access to only a (potentially large) list of the hypotheses examined during meta-training. A large gap indicates that any learner must extrapolate compositionally from the training hypotheses to solve the task; additionally, models can be compared to ideal learners that either do or do not engage in such extrapolation. We anticipate that this tool will be more broadly useful for analyzing other benchmarks with compositional splits.

**Models.** We evaluate models around various dimensions which concern the difficulty of learning productive concepts under uncertainty, including: 1) the modality in which the input is rendered (image, schemas, sounds), 2) method used for reasoning across objects in a scene (transformer,

**Types**

| | |
|---|---|
| P, Q | BOOL |
| A, B | STR |
| M, N | INT or FLOAT |
| SET[*] | A set of objects of any Type |

**Variables**

| | |
|---|---|
| x | Denotes an object in a scene |
| S | Denotes the SET of all objects in scene |
| S_{-x} | Denotes the SET of all objects except x |

**Constants (Illustrated for Images)**

| | |
|---|---|
| Counts | 1, 2, 3 |
| Materials | Rubber, Metal |
| Shapes | Cube, Sphere, Cylinder |
| Sizes | 0.35, 0.70 |
| X or Y location | 1, 2, 3, 4, 5, 6, 7, 8 (numbering starts from top-left of image) |

**Functions on Objects or Sets of Objects\***

*These compute property? (x). Illustration of properties in image domain:*

| | |
|---|---|
| size?(x) | Returns M, size of object x |
| material?(x) | Returns A, material of object x| |
| shape?(x) | Returns A, shape of object x |
| locationX? (x) | Returns INT M, x-coordinate of center of object x |
| locationY? (x) | Returns INT M, y-coordinate of center of object x |
| color? (x) | Returns A, color of object x |

*All these operations also apply when the argument is a SET.
Example color?(S) Returns  {color?(x): x in S}

**Quantifiers**

| | |
|---|---|
| for-all x in S | Returns TRUE if condition holds for all x |
| exists x in S | Returns TRUE if condition holds for any x |

**Logical Functions and Functions on BOOL Sets**

| | |
|---|---|
| P and Q | Returns TRUE if P and Q are both true |
| P or Q | Returns TRUE if either P or Q is true |
| not P | Returns TRUE iff P is False |
| all SET[P] | Returns TRUE if all elements of P are TRUE |
| any SET[P] | Returns TRUE if all elements of P are TRUE |
| count SET[P] | Returns INT, number of elements of P which are TRUE |

**Comparison Operations\***

| | |
|---|---|
| M = N | Returns TRUE if M and N are equal |
| A = B | Returns TRUE if A and B are equal |
| M > N | Returns TRUE if M is greater than N |
| M < N | Returns TRUE if M is lesser than M |

*All these operations also apply when one of the arguments is a SET.
Example SET[M] = N Returns  {M = N: M in SET[M]}

Figure 3: **Language of thought.** All valid (type-consistent) compositions of functions are potential complex concepts in our dataset. Note that the functions are illustrated for the case of images and schemas. Location, size, shape *etc.* correspond to different properties for sounds.

relation-network, global average pooling, concatenation), 3) whether or not training provides ground-truth symbolic descriptions of concepts, and 4) how negative examples are sampled. Overall, our evaluations suggest that there is substantial room for improvement in compositional reasoning under uncertainty, w.r.t the compositionality gap, representing a novel challenge for compositional learning.

**Summary of contributions:** 1) We introduce the Compositional Reasoning Under Uncertainty (CURI) benchmark for evaluating compositional, relational learning under uncertainty from observational data; 2) We introduce a 'compositionality gap' metric for measuring the difficulty of systematic generalization from train to test; 3) We provide various baseline models for benchmarking progess.

## 2 RELATED WORK

**Compositional Learning.** Related work has examined systematic generalization in pattern completion using Raven's matrices (PGM)  (Santoro et al., 2017; Hill et al., 2019) and visual question answering with CLEVR (Johnson et al., 2016; Bahdanau et al., 2019). CURI's use of the CLEVR renderer further invites particular comparison with that benchmark. Compared to these more deductive reasoning tests, CURI examines few-shot concept learning under substantial inherent uncertainty. Unlike puzzle solving or question answering, an ideal inductive learner on CURI cannot know the right rule with certainty. In essence, unlike CLEVR the "question" to be answered is not given to the model as input, but must be inferred – making the task more challenging. While PGMs do involve such an inference, once the constraints of a puzzle are identified, it does not: 1) have any uncertainty in the reasoning (which is crucial) and 2) involve any "concept" learning – where a concept applies to multiple images – as much as it involves "instance" matching to complete a sequence. In contrast, a successful CURI model behaves as if marginalizing over many hypotheses consistent with the observations e.g., (Tenenbaum & Griffiths, 2001; Xu & Tenenbaum, 2007; Piantadosi et al., 2016), an ability which is rarely studied directly in deep learning models (although see (Grant et al., 2019)).

Recently, Keysers et al. (2019) proposed a method to create "difficult" systematic splits based on the principle that they should share atoms but have maximally different compositions.  This is complementary to our splits, which provide interpretable notions of what each split tests such as disentangling, complexity, variable binding *etc.* Moreover, our variable binding split is predicated on having different atoms between train and test, and thus cannot be recovered by their methodology.

**Language of Thought (LOT).** Our choice of compositional concepts was most closely inspired by (Piantadosi et al., 2016) along with other studies of human concept learning in the Language of Thought (LOT) framework (Fodor, 1975; Goodman et al., 2008; Kemp & Jern, 2009; Piantadosi et al., 2012; Goodman et al., 2015; Overlan et al., 2017; Lake & Piantadosi, 2019). In typical LOT studies of human learning, the conceptual space $\mathcal{H}$ is defined through a probabilistic context-free grammar $\mathcal{G}$, which specifies a set of conceptual primitives and their rules of combination. Here, we use a LOT-inspired grammar $\mathcal{G}$ to generate an unbounded set concepts $\mathcal{H}$, while evaluating machine learning models trained without access to the underlying LOT.

## 3 COMPOSITIONAL REASONING UNDER UNCERTAINTY (CURI) DATASET

**Concept space.** The compositional concepts in CURI were inspired by the empirical and cognitive modeling work of Piantadosi et al. (2016).  The space of concepts (LOT) is defined by a context

free grammar ($\mathcal{G}$). Figure 3 shows the LOT and specifies how primitives and functions compose to produce a large unbounded concept space. The LOT has three variables: $\mathbf{x}$, representing an object in a scene, $S = \{\mathbf{x}\}_{i=1}^N$ representing the set of all objects in the scene, and $S_{-\mathbf{x}} = S/\{\mathbf{x}\}$, representing the set of all objects in the scene *except* $\mathbf{x}$. Each concept describes a rule composed of object and scene properties, logical operators, and/or comparison operators, and can be evaluated on a given scene $S$ to determine whether the scene satisfies the rule.

Object and scene properties are defined by functions which can be applied to objects or scenes: for example, `size?(`$\mathbf{x}$`)` yields the size of an object $\mathbf{x}$, while `size?(`$S$`)` returns a set with the sizes of all the objects ($\{$`size?(`$\mathbf{x}$`)` $: \mathbf{x} \in S\}$). Comparison and logical operators can be used to compare and relate various properties of objects in scenes. In contrast to Piantadosi et al. (2016), we include a `count` operator, which determines how many times a condition is satisfied by a set, which allows us to check how well deep learning models are able to count (Chattopadhyay et al., 2016; Johnson et al., 2016; Agrawal et al., 2017). Finally, quantifiers such as `exists` and `for-all` enrich the LOT by specifying the number of objects which must satisfy a given condition.

Consider the following example concept (Figure 1 bottom): "There exists a blue object in the scene and the rest of the objects are squares." To access the color of a given object, we use `color?(`$\mathbf{x}$`)` and to access the shape of a given object, we use `shape?(`$\mathbf{x}$`)`. To determine whether an object matches a specific property, we can combine this with equality: `shape?(`$\mathbf{x}$`)` `= "square"`. Finally, we can use `exists` to specify that at least one object must be blue, $S_{-\mathbf{x}}$ to specify all the objects except for that blue object, and `all` to specify that all the objects in $S_{-\mathbf{x}}$ must be squares. Putting it all together: `exists` $\mathbf{x} \in S$ `(color?(`$\mathbf{x}$`)` `= "blue")` `and all (shape?(`$S_{-\mathbf{x}}$`)` `= "square")`.

**Structured Generalization Splits.** A signature of productivity is the ability to handle novel combinations of known components (Fodor, 1975; Fodor & Pylyshyn, 1988). Thus, in CURI, we consider splits that require generalizing to novel combinations of known elements from our LOT (Figure 3), including combinations of constants, variables, and functions. We achieve this by creating disjoint splits of concepts $\mathcal{H}_{train}$ and $\mathcal{H}_{test}$ for training and evaluating models. By varying the held out elements and their combinations, we obtain splits that evaluate different axes of generalization. In practice, we use our grammar $\mathcal{G}$ to sample and filter a large set of concepts (see Appendix B.2 for more details), which yields a set of 14,929 concepts $\mathcal{H}$ for training and evaluation. We next describe how each split divides $\mathcal{H}$ into $\mathcal{H}_{train}$ and $\mathcal{H}_{test}$, to test productive, out of distribution generalization:

- **Instance IID**: Evaluates generalization to novel episodes from the same concept set. This is the standard setup in machine learning (Murphy, 2013), in which $\mathcal{H}_{train} = \mathcal{H}_{test}$. This is the only split where train and test concepts overlap.
- **Concept IID**: Evaluates generalization to novel concepts based on an arbitrary random split of the concepts into $\mathcal{H}_{train}$ and $\mathcal{H}_{test}$.[1]
- **Counting**: Evaluates the ability to learn a new concept $h$ with novel property-count combinations, e.g, the training concepts never filter for exactly '3 squares'.
- **Extrinsic properties**: Evaluates the ability to learn a new concept $h$, with novel combinations of extrinsic (e.g. location) and intrinsic (e.g. color) object properties.
- **Intrinsic properties**: Evaluates the ability to learn a new concept $h$ with novel combinations of intrinsic properties, e.g., the training concepts never reference both 'red' and 'rubber'.
- **Boolean operations**: Evaluates the ability to learn concepts which require application of a familiar boolean operation to a property to which the operation has never been applied previously.
- **Complexity split**: Evaluates generalization from simple concepts (those which have less than or equal to 10 symbols) to more complex concepts (longer than 10 symbols). This is indicative of the productivity (Fodor, 1975) exhibited by models, in generalizing from simpler concepts to more complex concepts.
- **Variable binding**: Evaluates learning of entirely novel intrinsic properties, e.g. the training concepts involve only "red", "blue", and "green" but test concepts involve "yellow" (although 'yellow' objects can still appear in training scenes). This is indicative of inferential coherence (Fodor, 1975) in models, in generalizing rules of inference to novel atoms.

A model that infers the underlying LOT during meta-training would be expected to perform well on any such systematic split. By comparing the performance of current models to to such ideal learners,

---

[1]While some strings $h$ might be different in surface form, they may yeild the same results when applied to images. In this split we account for such synonymy, and ensure that no two concepts which are synonyms are in different splits. See Appendix B.6 for more details.

this benchmark will allow us to evaluate progress on the systematic out-of-distribution generalization capabilities of our current models. Appendix C provides more details on the strucutred splits.

**From Concepts to Meta-learning Episodes.** A single episode comprises a support set ($D_{\text{supp}}$) and a query set ($D_{\text{query}}$), each of which is generated from a given concept, $h$. Formally, a support or query set $D$ has input data $\mathbf{u}$ and corresponding label $y$, *i.e.* $D = \{\{y_i\}_{i=1}^N, \{\mathbf{u}_i\}_{i=1}^N\}$. Each support and query set contains 5 positive and 20 negative examples — negative examples are oversampled since the space of negatives is generally much larger than that for positives. The set of positive examples are sampled uniformly from a categorical distribution over all positives. However, we consider two types of negatives: 1) easy negatives, in which the negatives are also sampled at random, and 2) hard negatives, in which negatives are generated from a closely related concept which also evaluates true on the positive examples in $D_{supp}$, such that these negatives are maximally confusing. Altogether, for each split, our train, validation, and test sets contain 500000, 5000, and 20000 episodes, respectively.

**Compositionality Gap.** A key aspect of our benchmark is to define the difficulty in learning that arises from the compositional structure of the concept space. Most of the splits above are structured in a way such that $\mathcal{H}_{\text{test}} \cap \mathcal{H}_{\text{train}} = \emptyset$ – forcing a learner to use the compositional structure of the concept space to generalize to $\mathcal{H}_{\text{test}}$. We conceptualize the difficulty of this task through the notion of its *compositionality gap*. Intuitively, the compositionality gap captures the difference between the generalization performance of an ideal compositional learner (*strong oracle*) compared to an ideal non-compositional learner that is unable to extrapolate outside the training concepts (*weak oracle*).

Formally, let $\Omega \in \{\text{strong}, \text{weak}\}$ denote an oracle over a concept space $\mathcal{H}_\Omega$. The posterior predictive distribution of an oracle for query scene $\mathbf{u}$ and query label $y \in \{0, 1\}$ is then given as: $p_\Omega(y|\mathbf{u}, D_{\text{supp}}) = \sum_{h \in \mathcal{H}_\Omega} p_\Omega(y|h, \mathbf{u}) p_\Omega(h|D_{\text{supp}})$, where $p_\Omega(h|D_{\text{supp}}) \propto p_\Omega(h) \, p(\{y_i\}_{i=1}^N | h; \{\mathbf{u}_i\}_{i=1}^N)$ and $p_\Omega(h)$ denote the posterior and prior, respectively. Given a metric of interest $M$ (e.g., mean average precision or accuracy), the compositionality gap of a learning task is then simply defined as the difference in performance of the strong and weak oracle when evaluating on concepts from $\mathcal{H}_{\text{test}}$, i.e., $M(p_{\text{strong}}) - M(p_{\text{weak}})$.

Using this notion of compositionality gap, we can then define ideal learners, i.e., the strong and weak oracle, simply via their priors. In particular, let $w(h)$ denote a weight on importance of each hypothesis[2] and let $\mathbf{I}$ denote the indicator function. We then define the prior of an oracle as $p_\Omega(h) = \sum_{h' \in \mathcal{H}_\Omega} w(h') \mathbf{I}[h' = h],$. The difference between strong and weak oracle lies in which concepts can be accessed in these priors.

In this formalism, the strong oracle has access to the union of train and test concepts; that is $\mathcal{H}_{\text{strong}} = \mathcal{H}_{\text{train}} \cup \mathcal{H}_{\text{test}}$. The weak oracle, on the other hand only assumes access to $\mathcal{H}_{\text{weak}} = \mathcal{H}_{\text{train}}$, which means it is unable to consider any hypothesis outside what has been seen in training and assigning it zero probability mass. Given a support set $D_{\text{supp}}$ this difference in priors leads then to different inferences on posteriors and allows us to quantify how compositionally novel a learning task is relative to these ideal learners.

## 4 METRICS AND BASELINES

During meta-test, given $D_{\text{supp}}$ models are evaluated on their ability to learn novel concepts. We use two metrics for quantifying this: 1) **Accuracy:** evaluates the accuracy of model predictions across the query set $D_{query}$, as is standard practice in meta-learning (Lake et al., 2019; Snell et al., 2017). Since there are more negative than positive labels, we report class balanced accuracy for better interpretability, averaging accuracies for the positive and negative query examples; and 2) **mean Average Precision (mAP):** evaluates models on a much larger number of test scenes $\mathcal{T}$ for each episode (comprising 44,787 scenes, 3 per each concept in $\mathcal{H}$). This resolves an issue that with a small query set, a strong model could achieve perfect accuracy without grasping the concept. Since episodes typically have many more negative than positive examples, Average Precision sweeps over different thresholds of a model's score and reports the average of the precision values at different recall rates, e.g., (Everingham et al., 2010). mAP is then the mean across all of the meta-test episodes.

---

[2]Set to log-linear in the prefix serialization length of hypothesis, inspired by the observation that longer hypotheses are more difficult for humans (Feldman, 2000). See Appendix B.3 for more details.

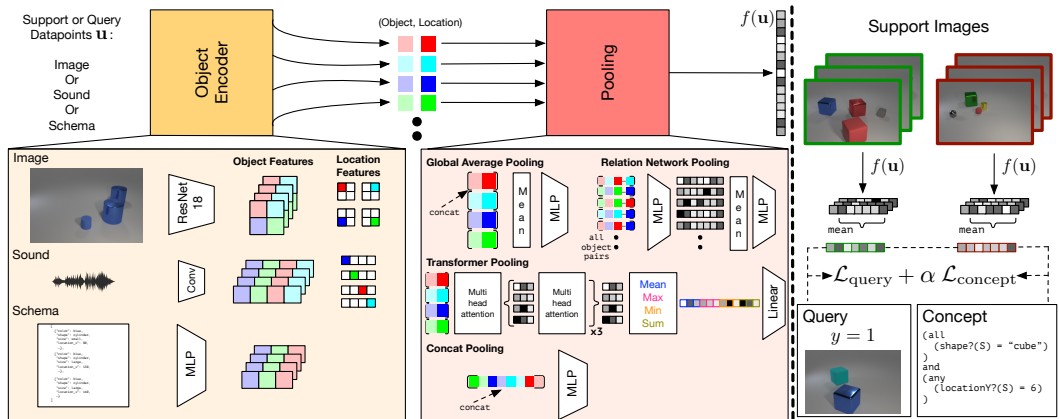

Figure 4: **Baseline models (left).** Different choices for the encoder $f(\mathbf{u})$ parameterization explored in the paper. We consider three modalities each of which is processed with a modality specific encoder, followed by four kinds of pooling architecture which take as input objects and their corresponding locations to provide an encoding for the datapoint. **Training (right).** The model is trained by processing the support images $D_{\text{supp}}$ with positive (green) and negative (red) images, using $f(\mathbf{x})$ to compute $\mathcal{L}_{\text{query}}$ which computes generalization error on queries and $\mathcal{L}_{\text{concept}}$ which learns to decode the true concept as an auxiliary task. Losses are weighted by $\alpha \geq 0$.

## 4.1 TRAINING LOSS

Denote by $\mathbf{u} \in \mathbb{R}^M$ the input to the model, which can be either in the form of image, sound or schema. We work in a binary classification setting with labels $y$ live in the space $\mathcal{Y} \in \{0, 1\}$. Then, given a support set $D_{\text{supp}} = \{\mathbf{u}_i, y_i\}_{i=1}^T$ and a query set $D_{\text{query}} = \{\mathbf{u}_i, y_i\}_{i=1}^T$, sampled in accordance with a productive concept $h$, our training objective for a single training instance can be written as $\mathcal{L}_{\text{query}} + \alpha\mathcal{L}_{\text{concept}}$. Here $\mathcal{L}_{\text{query}} = \sum_{\mathbf{u},y \in D_{\text{query}}} \log p(Y = y|\mathbf{u}, D_{\text{supp}})$ is a standard maximum likelihood meta-learning loss (Ravi & Larochelle, 2016; Snell et al., 2017; Finn et al., 2017), and $\mathcal{L}_{\text{concept}} = \log p(H = h|D_{\text{supp}})$ is an optional regularizer designed to encourage retaining information about the hypothesis of interest from the support set.

## 4.2 BASELINE MODEL ARCHITECTURES

Our baseline models (shown in Figure 4) parameterize the probability in the $\mathcal{L}_{\text{query}}$ term above using prototypical networks (Snell et al., 2017). The prototypical network consists of an embedding function $f = f_\theta$ and uses it to compute prototypes $\mathbf{c}_p$ and $\mathbf{c}_n$ for positive and negative examples by averaging $f(\mathbf{u})$ for positive and negative examples in the support set respectively. In equations, given a query datapoint $\mathbf{u}'$, we compute

$$p(Y = y|\mathbf{u}; D_{\text{supp}}) = \frac{exp(||f(\mathbf{u}') - \mathbf{c}_p||^2)}{exp(||f(\mathbf{u}') - \mathbf{c}_p||^2) + exp(||f(\mathbf{u}') - \mathbf{c}_n||^2)} \tag{1}$$

In this formalism, the models we study in this paper span different choices for $f$. Roughly, in each modality, we start with an encoder that converts the raw input into a set of vectors, and then a pooling operation that converts that set of vectors into a single vector. In the case of images and sound (input as spectrograms), the encoder is a ResNet-18; and the set of vectors is a subsampling of spatial locations; and for schemas we vectorize components with a lookup table and combine them into a set via feed-forward networks. In the case of images and sounds, the output of the encoder is enriched with position vectors. For the pooling operation, we study global averaging, concatenation, relation networks (Santoro et al., 2017) and transformers (Vaswani et al., 2017) equipped with different pooling operations (max, mean, sum, min) for reasoning inspired by Wang et al. (2019) (Figure 4 middle panel, also see Appendix F for more details).

For the probability in $\mathcal{L}_{\text{concept}}$, we represent the concept as a sequence by prefix serialization and then use an LSTM (Hochreiter & Schmidhuber, 1997) to parameterize $p(h|D_{\text{supp}}) = \Pi_{s=1}^S p(h_s|h_{1\cdots t-1}; D_{supp})$. At each step of the LSTM we concatenate $[\mathbf{c}_p, \mathbf{c}_n]$ to the input.

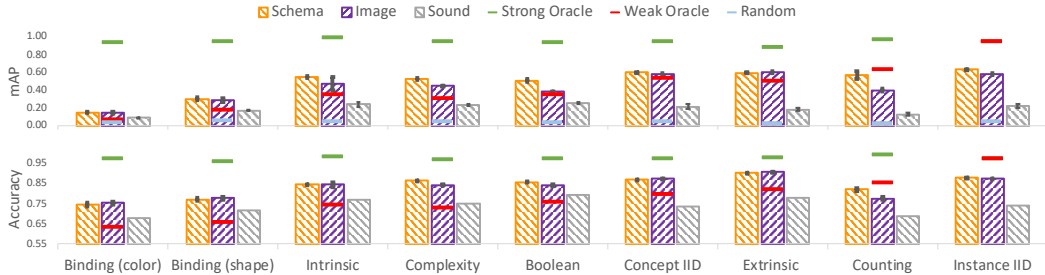

Figure 5: **Compositionality Gap**. Different splits (x-axis) plotted w.r.t performance of the strong oracle (green line) and weak oracle (red line) on the mAP (top) and Accuracy (bottom) evaluated on respective test splits (using hard negatives in support and query sets). Difference between the two is the compositionality gap (`comp gap`). **Yellow:** shows the (best) relation-net model on schema inputs, **purple:** shows the model on image inputs, and **gray:** shows the model on sound inputs. Error bars are `std` across 3 independent model runs.

## 5 EXPERIMENTAL RESULTS

We first discuss the compositionality gap induced by the different generalization splits and then delve into the impact of modeling choices on performance on the generalization splits. All models are trained for 1 million steps, and are run with with 3 independent training runs to report standard deviations. We sweep over 3 modalities (image, schema, sound), 4 pooling schemes (avg-pool, concat, relation-net, transformer), 2 choices of negatives (hard negatives, random negatives) and choice of language ($\alpha = 0.0, 1.0$). Unless mentioned otherwise in the main paper we focus on results with hard negatives and $\alpha = 0.0$. When instantiated for a given modality, we note that the encoders $f(\mathbf{u})$ (Figure 4) all have a similar number of parameters. The appendix contains details of the exact hyperparameters (Appendix E), and more comprehensive results for each split (Appendix G.7).

### 5.1 DATASET DESIGN AND COMPOSITIONALITY

**How compositional are the structured splits?**. Our main results are shown in Figure 5. Using our model-independent measure of the compositionality gap (Section 3), different splits present varying challenges for generalizing from train to test. The most difficult splits, with the largest compositionality gaps, are the Binding (color) and Binding (shape), which is reasonable since they require learning concepts with entirely new property-values. In contrast, the easiest split with the smallest compositionality gaps is the Instance IID split since it does not require compositionality. Finally, while the mAP metric exposes a larger value of `comp gap`, the ordering of splits in terms of `comp gap` is same for both metrics – suggesting similar coarse-grained notions of compositionality.

Results for the best overall architecture, a relation network (relation-net), is shown in Figure 5. Network performance on the easiest data format (schema; yellow bars) is generally better than the weak oracle, but substantially worse than the strong oracle. Counting is a particularly challenging split where the models underperform even the weak oracle. Broadly, this suggests that the models capture some notion of compositionality – especially for images and schemas – relative to a weak oracle that rigidly considers only training hypotheses, but there is substantial room to improve (especially with respect to the more stringent mAP metric). These results demonstrate that CURI provides a challenging yet tractable setting for evaluating the compositional capabilities of models.

Finally, we found that the performance on the Instance IID split is not equal to the weak (and strong) oracle—which are both equal in this case—indicating that the best model does not make ideal posterior predictions even when compositionality is not an issue. Ideal predictions in this case would require the network to behave as if marginalizing over the training hypotheses, as the strong oracle does. A similar plot to Figure 5 can be found in Appendix G.4 for random negatives.

**Influence of Negatives**. Previous work (Hill et al., 2019) has shown that the choice of random *v.s.* hard negatives for training and evaluation impacts compositional generalization substantially in the case of a particular set of analogical reasoning models. However, we argue that such decisions on dataset design can be made more objectively if one can evaluate the model-independent `comp gap`. In our context, we find that the `comp gap` with mAP when using random negatives decreases on average by $5.5 \pm 1.4\%$ compared to when we use hard negatives. This indicates that it is not only the choice of $\mathcal{H}_{train}$ and $\mathcal{H}_{test}$, which are identical for a given compositional split (say Counting),

but also the choice of the negatives which "makes" the task compositionally novel. More generally, this indicates that the `comp gap` has utility as a more general diagnostic tool for making principled design decisions in compositional learning settings *without* the confound of specific model decisions.

## 5.2 Differences Between Models

**Best Models**. In general, the best performing model is the relation-net applied to schema inputs, outperforming other combinations of models and input modalities on the Boolean, Concept IID, Complexity, and Instance IID splits on both the mAP as well as accuracy metrics (Figure 5); although as mentioned above, none of the models are close to the strong oracle. It is closely followed by the transformer model on schema inputs, which performs the best on Binding (color), Binding (shape), and Intrinsic splits (Appendix G.7). Utilizing schema inputs proves easier for abstraction except for the Extrinsic setting, where the task requires generalization to novel locations for objects in images, which is well supported by the inductive bias of the CNN encoder (Figure 4). In this case, the image-transformer gets an mAP of $62.1 \pm 0.7\%$, compared to the next best schema-transformer model at $60.9 \pm 0.7$. Further, relational learning proves more crucial in the schema case than for images, with all image models (regardless of pooling) performing better than $59.4 \pm 1.3\%$ mAP (achieved for image-avg-pool) while schema-avg-pool models get only get to $53.4 \pm 1.5\%$.

**When to use a transformer?** Transformer models appear to outperform relation networks in splits concerning disentangling. For instance, for the Intrinsic split with schema-relation-net is at $55.1 \pm 0.8\%$ *v.s.* $57.9 \pm 0.6\%$ for schema-transformer. Similarly, for the Extrinsic split the image-transformer is at $62.1 \pm 0.7\%$ compared to the image-relation-net at $60.8 \pm 1.1\%$. We hypothesize that this is because the iterative message passing via. attention in transformers improves object representations for disentangling compared to relation networks that lack such a mechanism.

**What is the relative difficulty of abstraction from different modalities?** One of the key contributions of our work is in providing multiple modalities (image, schema, sound) for productive concept learning. We next characterize the difficulty of abstraction based on modality for the various generalization settings. In the Intrinsic setting, we find that the schema models, which have access to a "perfect" disentangled representation significantly outperform image models—a schema-avg-pool model gets an mAP of $52.7 \pm 3.1\%$ while an image-avg-pool model gets to $34.4 \pm 0.0\%$ mAP.

Similarly, for the Counting split where the total number of objects are exactly specified in the schema (Figure 4), schemas are substantially better than images. For example, schema-relation-nets get to $56.25 \pm 5.32\%$ mAP while image-avg-pool is at $48.4 \pm 1.2\%$ mAP. Interestingly, the next best model—image-relation-net—is substantially worse, at $39.45 \pm 1.6\%$. Curiously, for while transformer models perform well at disentangling, they seem to be quite poor for Counting, with image-transformer models getting to only $32.4 \pm 1.4\%$ mAP, suggesting a potential weakness for transformers. Overall, there appears to be an intimate link between the generalization setting and the input modality, suggesting avenues where representation learning could be improved for a given modality (e.g. images), relative to the kind of reasoning one is interested in (e.g. counting).

**When does language help?** On average, training models with explicit concept supervision using the concept loss (Section 4.1) improves performance by $2.8 \pm 0.6\%$ mAP (SEM error). This is a small boost relative to the gap between the original model and the strong oracle, suggesting that this simple auxiliary loss is not sufficient to internalize the LOT in a neural network. Overall, image models benefit more from language than schema models which natively utilize symbols (Appendix G.3).

## 6 Conclusion

We introduced the compositional reasoning under uncertainty (CURI) benchmark for evaluating few-shot concept learning in a large compositional space, capturing the kinds of productivity, unboundness and underdetermination that characterize human conceptual reasoning. We instantiate a series of meta-learning tasks, and evaluate numerous baseline models on various aspects of compositional reasoning under uncertainty, including inferential coherence, boolean operation learning, counting, disentangling, *etc.* Further, we introduce the notion of a compositionality gap to quantify the difficultly of each generalization type, and to estimate the degree of compositionality in current deep learning models. We hope our contributions of dataset, compositionality gaps, evaluation metrics and baseline models help spur progress in the important research direction of productive concept learning.

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

## A    EXAMPLE EPISODES FROM THE DATASET

We show examples from the Concept IID split test set comprising the ground truth productive concept (top), along with the support and query sets for meta learning (rendered as images), the alternate hypotheses which are consistent with the support set – that is, other hypotheses which could also have generated the positive and negative examples in the support set – and the concepts based on which we pick the hard negatives Figures 6 to 11.

## B    ADDITIONAL DATASET DETAILS

We first provide more details of the concept space $\mathcal{G}$, then explain how we obtain $\mathcal{H}$, the space of concepts for training and evaluation, provide more details of the structured splits, and finally explain the weight $w(h)$ based on which we sample concepts.

### B.1    MORE DETAILS OF THE GRAMMAR

We provide below the full grammar used to sample concepts, where $A \rightarrow B|C$ means that $A$ can expand to either $B$ or $C$ under the rules defined by the grammar. We always start expanding at the START token and then follow the rules of the grammar until we hit a terminal node (which does not have any expansions defined). As and where possible, we followed the insights from Piantadosi et al. (2016) in choosing the sampling probabilities for various completions based on how well humans seem to be able to learn the corresponding primitive. For example, we sample utterances with disjunctions (or) less frequently since they are known to be difficult for humans to learn. Based on Kemp & Jern (2009), we chose to represent location as a discrete entity, such that relative, and categorical notions of left or right simply become comparisons in the location space (location? $\mathbf{x} >$ location? $S_{-\mathbf{x}}$), unlike the CLEVR dataset (Johnson et al., 2016) which defines categorical relational objects.

Here is the full grammar $\mathcal{G}$ used for sampling the concepts (as explained in the main paper, $S_{-\mathbf{x}} = S/\{\mathbf{x}\}$). Note that the grammar always generates strings in postfix notation and thus the operands in each expansion occur before the operation:

Productive Concept: for-all x in S and(>(8, locationX?( x ) ), =(purple, color?( x ) ) )

Valid Hypotheses

* for-all x in S =(purple, color?( x ) ) {log-prob [-0.852]}
* for-all x in S =(color?( x ), purple ) {log-prob [-0.852]}
* all(color?( S ), purple ) {log-prob [-1.950]}
* exists x in S and(=(metal, material?( x ) ), all(color?( S ), purple ) ) {log-prob [-8.523]}
* for-all x in S and(=(color?( x ), purple ), any(material?( S ), metal ) ) {log-prob [-8.523]}

Hypotheses for Hard Negatives

* for-all x in S or(=(shape?( x ), cube ), all(color?( S ), purple ) )
* for-all x in S or(=(purple, color?( x ) ), =(sphere, shape?( x ) ) )
* exists x in S and(any(color?( S_{-x} ), purple ), all(color?( S_{-x} ), purple ) )
* for-all x in S or(=(shape?( x ), sphere ), =(purple, color?( x ) ) )
* for-all x in S or(=(purple, color?( x ) ), =(shape?( x ), cube ) )

Figure 6: Qualitative Example of an Episode in CURI dataset. Best viewed zooming in, in color.

Productive Concept: exists x in S and(all(locationY?( S_{-x} ), 7 ), any(color?( S_{-x} ), red ) )

Valid Hypotheses

* any(locationY?( S ), 7 ) {log-prob [-1.621]}
* exists x in S >(locationY?( x ), 6 ) {log-prob [-1.621]}
* exists x in S any(locationY?( S_{-x} ), 7 ) {log-prob [-1.621]}
* exists x in S all(locationY?( S_{-x} ), 7 ) {log-prob [-1.647]}
* exists x in S =(locationY?( x ), 7 ) {log-prob [-2.314]}

Hypotheses for Hard Negatives

* exists x in S and(all(color?( S_{-x} ), red ), >(8, locationX?( x ) ) )
* exists x in S and(all(color?( S_{-x} ), red ), >(7, locationX?( x ) ) )
* exists x in S and(all(color?( S_{-x} ), red ), any(size?( S_{-x} ), 0.35 ) )
* exists x in S and(>(8, locationY?( x ) ), >(locationY?( x ), 5 ) )
* exists x in S or(all(shape?( S ), sphere ), all(color?( S_{-x} ), red ) )

Figure 7: Qualitative Example of an Episode in CURI dataset. Best viewed zooming in, in color.

Productive Concept: for-all x in S and(>(8, locationX?( x ) ), =(shape?( x ), sphere ) )

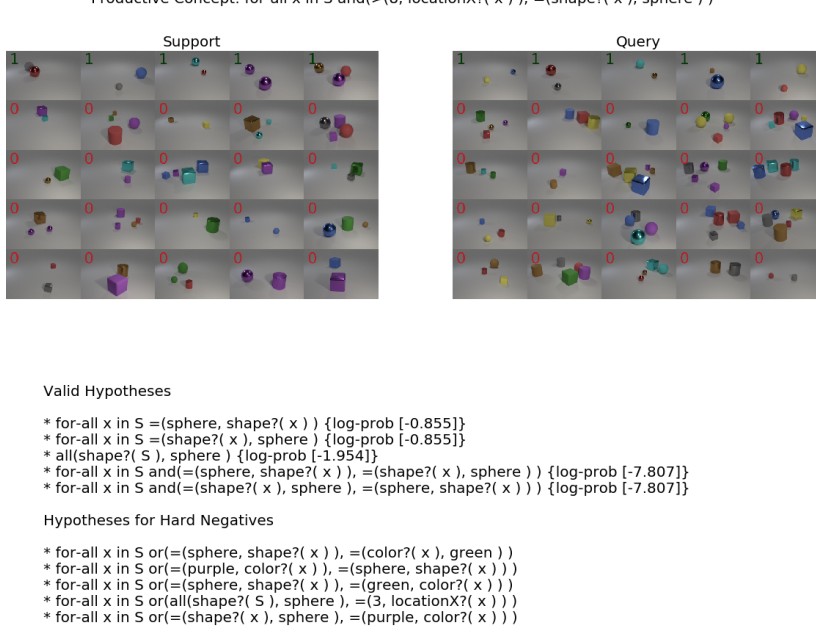

Valid Hypotheses

* for-all x in S =(sphere, shape?( x ) ) {log-prob [-0.855]}
* for-all x in S =(shape?( x ), sphere ) {log-prob [-0.855]}
* all(shape?( S ), sphere ) {log-prob [-1.954]}
* for-all x in S and(=(sphere, shape?( x ) ), =(shape?( x ), sphere ) ) {log-prob [-7.807]}
* for-all x in S and(=(shape?( x ), sphere ), =(sphere, shape?( x ) ) ) {log-prob [-7.807]}

Hypotheses for Hard Negatives

* for-all x in S or(=(sphere, shape?( x ) ), =(color?( x ), green ) )
* for-all x in S or(=(purple, color?( x ) ), =(sphere, shape?( x ) ) )
* for-all x in S or(=(sphere, shape?( x ) ), =(green, color?( x ) ) )
* for-all x in S or(all(shape?( S ), sphere ), =(3, locationX?( x ) ) )
* for-all x in S or(=(shape?( x ), sphere ), =(purple, color?( x ) ) )

Figure 8: Qualitative Example of an Episode in CURI dataset. Best viewed zooming in, in color.

Productive Concept: exists x in S and(all(locationY?( S_{-x} ), 6 ), any(size?( S_{-x} ), 0.7 ) )

Support          Query

Valid Hypotheses

* exists x in S all(locationY?( S_{-x} ), 6 ) {log-prob [-0.008]}
* exists x in S >(count=(locationY?( S_{-x} ), 6 ), count=(size?( S_{-x} ), 0.35 ) ) {log-prob [-6.668]}
* for-all x in S =(count=(locationY?( S ), 6 ), count=(shape?( S ), shape?( x ) ) ) {log-prob [-6.690]}
* for-all x in S =(count=(shape?( S ), shape?( x ) ), count=(locationY?( S ), 6 ) ) {log-prob [-6.690]}
* exists x in S and(>(locationY?( x ), 5 ), >(size?( x ), 0.35 ) ) {log-prob [-7.451]}

Hypotheses for Hard Negatives

* and(any(size?( S ), 0.7 ), any(locationY?( S ), 6 ) )
* exists x in S and(any(locationY?( S ), 6 ), >(locationY?( x ), locationX?( x ) ) )
* exists x in S and(=(locationY?( x ), 6 ), any(shape?( S ), sphere ) )
* exists x in S =(count=(locationY?( S ), 6 ), count=(locationY?( S ), locationY?( x ) ) )
* exists x in S >(count=(locationY?( S_{-x} ), 6 ), count=(locationY?( S ), 2 ) )

Figure 9: Qualitative Example of an Episode in CURI dataset. Best viewed zooming in, in color.

Productive Concept: exists x in S and(all(material?( S_{-x} ), rubber ), all(color?( S_{-x} ), cyan ) )

Valid Hypotheses

* exists x in S and(any(material?( S_{-x} ), rubber ), all(color?( S_{-x} ), cyan ) ) {log-prob [-0.983]}
* exists x in S and(all(color?( S_{-x} ), cyan ), all(material?( S_{-x} ), rubber ) ) {log-prob [-1.162]}
* exists x in S and(all(material?( S_{-x} ), rubber ), all(color?( S_{-x} ), cyan ) ) {log-prob [-1.162]}

Hypotheses for Hard Negatives

* >(count=(color?( S ), cyan ), count=(material?( S ), metal ) )
* exists x in S and(all(color?( S_{-x} ), cyan ), any(locationY?( S ), 3 ) )
* exists x in S all(color?( S_{-x} ), cyan )
* exists x in S and(all(color?( S_{-x} ), cyan ), any(color?( S_{-x} ), cyan ) )
* exists x in S and(all(color?( S_{-x} ), cyan ), any(size?( S ), 0.7 ) )

Figure 10: Qualitative Example of an Episode in CURI dataset. Best viewed zooming in, in color.

Productive Concept: exists x in S and(all(locationY?( S_{-x} ), 5 ), >(locationY?( x ), locationX?( x ) ) )

Valid Hypotheses

* exists x in S and(all(locationY?( S_{-x} ), 5 ), >(locationY?( x ), locationX?( x ) ) ) {log-prob [0.000]}

Hypotheses for Hard Negatives

* for-all x in S or(all(locationX?( S ), 1 ), >(locationY?( x ), locationX?( x ) ) )
* for-all x in S and(>(locationY?( x ), locationX?( x ) ), >(8, locationY?( x ) ) )
* for-all x in S or(>(locationY?( x ), locationX?( x ) ), =(color?( x ), brown ) )
* for-all x in S or(=(yellow, color?( x ) ), >(locationY?( x ), locationX?( x ) ) )
* for-all x in S or(=(purple, color?( x ) ), >(locationY?( x ), locationX?( x ) ) )

Figure 11: Qualitative Example of an Episode in CURI dataset. Best viewed zooming in, in color.

```
START  -> λ S. BOOL  exists=  |  λ S. BOOL  for-all=

BOOL   -> BOOL BOOL  and  |  BOOL BOOL  or  |  BOOL  not  |
   C C =  |  SH SH =  |  M M =  |  SI  SI  =  |  L L =  |
   NUM NUM =  |  SI  SI  >  |  L L  >  |  NUM NUM  >  |
   SETFC C  all  |  SETFSH SH  all  |  SETFM M  all  |
   SETFSI SI  all  |  SETFL L  all  |  SETFC C  any  |
   SETFSH SH  any  |  SETFM M  any  |  SETFSI SI  any  |
   SETFL L  any

NUM    -> SETFC C  count=  |  SETFSH SH  count=  |
   SETFM M  count=  |  SETFSI SI  count=  |  SETFL L  count=
NUM    -> 1  |  2  |  3

SETFC -> SET  FC
SETFSH-> SET  FSH
SETFM -> SET  FM
SETFSI-> SET  FSI
SETFL -> SET  FL

C       -> gray  |  red  |  blue  |  green  |  brown  |  purple  |
   cyan  |  yellow
C       -> OBJECT  FC

SH      -> cube  |  sphere  |  cylinder
SH      -> OBJECT  FSH

M       -> rubber  |  metal
M       -> OBJECT  FM

SI      -> large  |  small
SI      -> OBJECT  FSI

L       -> 1  |  2  |  3  |  4  |  5  |  6  |  7  |  8
L       -> OBJECT  FL

FC      -> color?
FSH     -> shape?
FM      -> material?
FSI     -> size?
FL      -> locationX?  |  locationY?

OBJECT-> x
SET:  S  |  S_{-x}
```

## B.2  SAMPLING

We sample 2000000 initial hypotheses from the CFG $\mathcal{G}$, and impose a maximum depth in the recursion tree of 6 when sampling. That is, no node has a depth larger than 6 in the recursion through which we generate concepts from the grammar $\mathcal{G}$. We then reject and filter the hypotheses to obtain a set of "interesting" hypotheses $\mathcal{H}$ used in the main paper explained in more detail below:

**Rejection Sampling:** We reject the following string combinations after sampling from the grammar $\mathcal{G}$:

- All programs which contain "$\lambda S.$ for-all x" and "$S_{-x}$" in the same program. This is asking that for all objects in a scene, a certain property is satisfied by everything other than the object, which is the same as saying, for all objects in the scene.

- All programs where we compare the properties of the same object to itself, e.g. `color?(x) == color?(x)`, where color? can be any function applied to the object.

- All programs where we have the following string: `exists(color?(S) == color?(x))` or `for-all(color?(S) == color?(x))` where color? can be any function applied to the object.

- All programs which evaluate to true on schemas more than 10% of the time and less than 10 times. The former condition ensures that we work with concepts which are in some sense interesting and surprising (as opposed to concepts which are always trivially true), and the second condition ensures that we have unique schmeas or datapoints to place in the support and query sets, which both have 5 positive images each.

We provide examples of concepts which get rejected for being true too often below:

```
exists=x \in S or(
  =(locationX?( x ), locationY?( x ) ),
  any(color?( S ), brown )
)
exists=x \in S and(
  exists=(locationY?( S ), locationX?( x ) ),
  any(color?( S ), brown )
)
exists=x \in S or(
  all(color?( S ), gray ),
  all(color?( S ), brown )
)
```

See Appendix C for more details on the structured generalization splits which yeild train concepts $\mathcal{H}_{\text{train}}$ and test concepts $\mathcal{H}_{\text{test}}$.

## B.3 CONCEPT PRIOR WEIGHT $w(h)$

We next explain the form of the prior weight $w(h)$ that we use for defining the prior over the concepts provided to the models (both oracles as well as deep learning models). Given $l(h)$, the number of tokens in the postfix serialization of the concept $h$, the unnormalized weight $\tilde{w}(h)$ is log-linear in the length, and is defined as follows:

$$\tilde{w}(h) \propto \exp -0.2 \cdot l(h) \tag{2}$$

Given a split $\Omega \in \{train, test\}$, the final, normalized weight is given as:

$$w(h) = \frac{\tilde{w}(h)}{\sum_{\mathcal{H}_\Omega} \tilde{w}(h)} \tag{3}$$

As explained in the main paper, the final prior for a hypothesis given a split $\Omega$ is $p(h) = \sum_{h' \in \mathcal{H}_\Omega} w(h)\mathbf{I}[h = h']$.

Our choice of the log-linear weight is inspired by the observation in cognitive science that longer boolean concepts are harder for people to learn (Feldman, 2000).

Here are some examples of hypotheses with a high weight (computed on $\Omega = $ train $\cup$ test):

```
exists x in S =(2, count=(color?( $S_{-x}$ ), cyan ) ),
exists x in S >(locationY?( x ), 6 ),
=(count=(color?( S ), brown ), 3 ),
>(count=(locationX?( S ), 3 ), 2 ),
any(locationY?( S ), 6 ),
=(1, count=(locationY?( S ), 7 ) ),
=(3, count=(locationY?( S ), 3 ) ),
all(locationX?( S ), 2 ),
exists x in S all(locationY?( $S_{-x}$ ), 5 ),
```

```
=(2,  count =( color ?(  S  ),  blue  )  ),
for−all  x  in  S  not (  >(6,  locationX ?(  x  )  )  ),
=( count =( color ?(  S  ),  gray  ),  2  ),
=(2,  count =( color ?(  S  ),  gray  )  )
```

## B.4  EXECUTION ON IMAGES.

In order to create the perceptual inputs in the dataset $\mathcal{U}$, we sample images using the renderer for CLEVR from Johnson et al. (2016), changing the range of objects to $[2, 5]$, to reduce clutter and enable easier learning of predicates like `any` and `all` for models. [3] The CLEVR renderer produces scenes $\mathbf{u}$ with pixels as well as an associated schema file $\mathbf{u}^s$ detailing the properties of all the objects sampled in the scene, including their location, shape, size, material, and rotation. Based on this, we convert our sampled concepts into postfix notation and execute them on the schemas using an operator stack. Concretely, execution of the concept $h \in \mathcal{H}$ on $\mathbf{u}^s$ yields a boolean true or false value $\{0, 1\}$. We execute each such hypothesis on a set of 990K images, yielding scores of how often a hypothesis is true for an image. We threshold this score to retain the subset of hypotheses which are true for no more than 10% of the images and are true at least for 10 images, to pick a subset of "interesting" hypotheses $\mathcal{H}'$ for training models.

**Bias.** The image dataset here sampled itself has a bias in terms of the location coordinates (in the pixel space). The CLEVR dataset generation process samples objects in the 3d (top-down x, y) coordinate space uniformly (from a grid of -3, to +3). However, since the camera is always looking into the scene from outside, the image formation geometry implies that in the camera / image coordinates most of the objects appear to be away from the scene and very few are close to the camera. Thus, in terms of the y-coordinates we observe in the image coordinates a bias in terms of the distribution not being unifrom. This also makes sense in general, as even in the real world, objects are not found very close to the camera or very far away from the camera in general. See Figure 12 for all the biases in the observation space $\mathbf{u}$ computed over 990K sampled images.

## B.5  AUDIO.

To build the audio data, we use clips of orchestral instruments playing various pitches downloaded from `https://philharmonia.co.uk/resources/sound-samples/`. We make the following mappings of object properties:

- $x$ location $\rightarrow$ temporal location. larger $x$ bin means the note is played later.
- $y$ location $\rightarrow$ pitch. All pitches between the instruments are the same (up to octaves).
- color $\rightarrow$ instrument
    - gray $\rightarrow$ trumpet
    - red $\rightarrow$ clarinet
    - blue $\rightarrow$ violin
    - green $\rightarrow$ flute
    - brown $\rightarrow$ oboe
    - purple $\rightarrow$ saxaphone
    - cyan $\rightarrow$ french-horn
    - yellow $\rightarrow$ guitar
- shape $\rightarrow$ amplitude profile; either getting louder, getting softer, or constant volume
- size $\rightarrow$ total volume
- material $\rightarrow$ low-pass filtering or no filtering.

All binned quantities use the same number of bins as in the image domain.

---

[3]Since the chances of a constraint being true for all obejcts reduce exponentially as the number of objects increases.

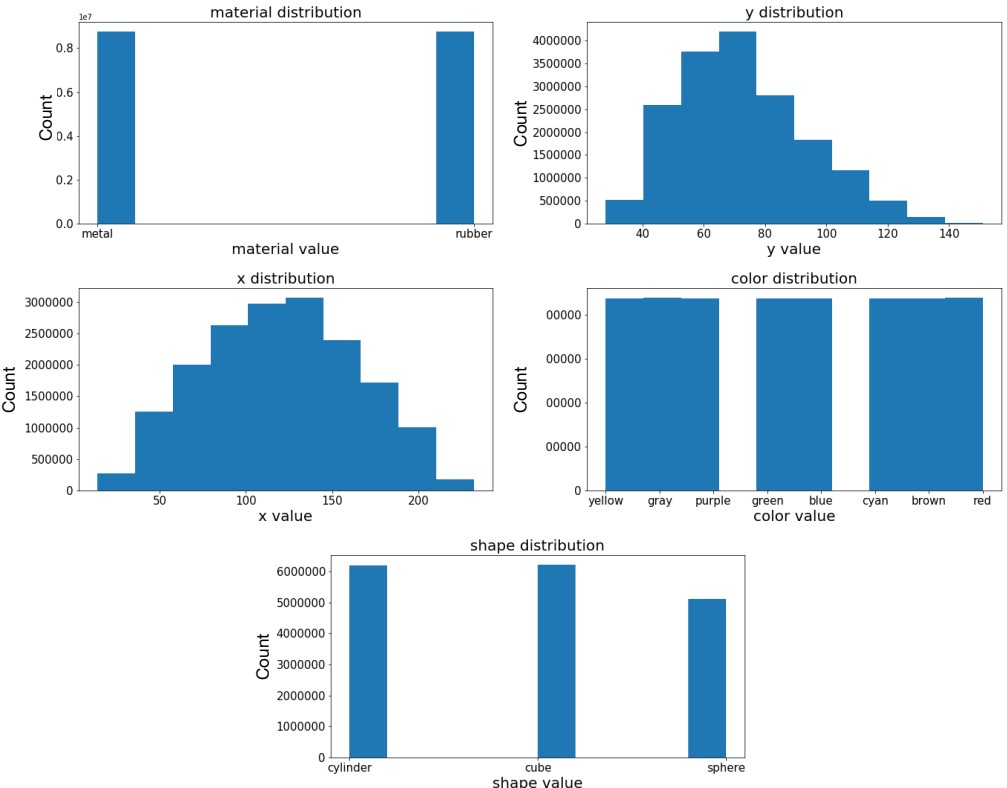

Figure 12: Histogram of properties found in inputs **u** in the dataset. We notice that the properties are all largely uniform with bias in x and y-coordinates towards the center of the image.

### B.6 ANALYSIS OF SYNONOMY OF CONCEPTS.

We next show an analysis of concepts which have the same evaluation signatures on a large set of 990K images, and are thus synonymous (in context of the dataset at hand). Note that while some of these concepts might be truly synonymous to each other (for example, $A > B$ is the same as $B < A$), others might be synonymous in context of the image distribution we work with. For example, size can never be greater than 0.7 in our dataset and location can never be greater than 8, and thus asking if location is greater than 8 or size is greater than 0.7 has the same semantics on our dataset. In Figure 13 we show each such "concept" or "meaning", which is a cluster of hypotheses which all evaluate to the same truth value and plot a histogram of how many hypotheses each cluster tends to have. We notice that most of the concepts have 1 synonym (i.e. there is only one concept with the particular) evaluation signature, with a long tail going upto 80 synonyms in a concept. In the Concept IID split we ensure that none of the concepts which have the same signature are found across the train/val/test splits.

## C DETAILED DISCUSSION OF THE STRUCTURED SPLITS

We provide more details on how each of the structured splits described in Sec. 3 of the main paper are created. Assuming access to $\mathcal{H}$, the space of concepts sampled and filtered from the grammar $\mathcal{G}$, we use various heuristics to produce the generalization splits in the paper:

- Instance IID: This split is trivial since $\mathcal{H}_{\text{train}} = \mathcal{H}_{\text{test}} = \mathcal{H}$

- Concept IID: This split divides concepts into train and test by picking concepts at random from $\mathcal{H}$ and assigning them to $\mathcal{H}_{\text{train}}$ or $\mathcal{H}_{\text{test}}$ while ensuring that no two concepts which are synonyms Appendix B.6 are found in different splits.

- Boolean: This split forms cross product of all possible colors and {and, or} boolean operators, and partitions a subset of such combinations which we want to only occur in test. We use the following tokens for test:

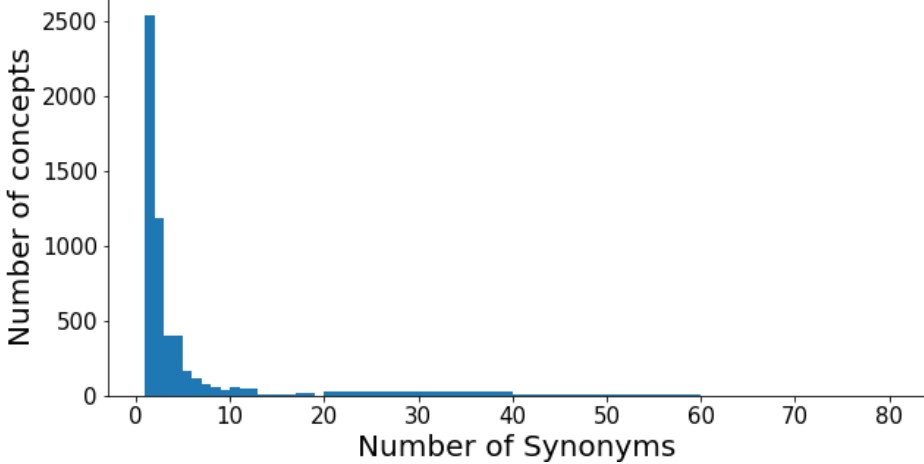

Figure 13: Histogram of number hypotheses with same evaluation signatures.

```
'green', 'or' | 'purple', 'and' | 'cyan', 'and' |
'red', 'or' | 'green', 'and'
```

We then create $\mathcal{H}_{\text{test}}$ to contain all concepts which have any of the combinations above. For example, if a concept has both green and or we would place it in $\mathcal{H}_{\text{test}}$. After every feasible candidate is placed in $\mathcal{H}_{\text{test}}$ based on this heurisitc, the remaining concepts in $\mathcal{H}$ are assigned to $\mathcal{H}_{\text{train}}$.

- Extrinsic: This split forms cross product of all possible colors and locations in the dataset, and partitions a subset of such combinations that we only want to occur in test. We use the following tokens for test (only a subset shown for illustration):

```
'7', 'gray' | '1', 'red' | '3', 'purple' |
'1', 'blue' | '8', 'cyan' | '5', 'yellow' |
'5', 'green' | '3', 'yellow' | '7', 'purple' |
'2', 'blue' | '3', 'cyan'
```

We then create $\mathcal{H}_{\text{test}}$ to contain all concepts which have any of the combinations above. For example, if a concept has both gray and 7, and is related to location, that is contains locationX? or locationY? keywords, we would place it in $\mathcal{H}_{\text{test}}$. After every feasible candidate is placed in $\mathcal{H}_{\text{test}}$ based on this heurisitc, the remaining concepts in $\mathcal{H}$ are assigned to $\mathcal{H}_{\text{train}}$.

- Intrinsic: This split forms cross product of all possible colors and materials in the dataset, and partitions a subset of such combinations that we only want to occur in test. We use the following tokens for test:

```
'green', 'metal' | 'purple', 'rubber' | 'cyan', 'rubber' |
'red', 'metal' | 'green', 'rubber'
```

We then create $\mathcal{H}_{\text{test}}$ to contain all concepts which have any of the combinations above. For example, if a concept has both green and metal, and is related to material, that is contains material? keyword, we would place it in $\mathcal{H}_{\text{test}}$. After every feasible candidate is placed in $\mathcal{H}_{\text{test}}$ based on this heurisitc, the remaining concepts in $\mathcal{H}$ are assigned to $\mathcal{H}_{\text{train}}$.

- Binding (color): This split takes all possible colors in the dataset, and partitions a subset of colors that we only want to occur in test. We use the following tokens for test:

```
'purple' | 'cyan' | 'yellow'
```

We then create $\mathcal{H}_{\text{test}}$ to contain all concepts which have any of the tokens above. For example, if a concept has purple, we would place it in $\mathcal{H}_{\text{test}}$. After every feasible candidate is placed in $\mathcal{H}_{\text{test}}$ based on this heurisitc, the remaining concepts in $\mathcal{H}$ are assigned to $\mathcal{H}_{\text{train}}$.

- Binding (shape): This split takes all possible shapes in the dataset, and partitions a subset of shapes that we only want to occur in test. We use the following tokens for test:

    'cylinder'

    We then create $\mathcal{H}_{\text{test}}$ to contain all concepts which have any of the tokens above. For example, if a concept has cylinder, we would place it in $\mathcal{H}_{\text{test}}$. After every feasible candidate is placed in $\mathcal{H}_{\text{test}}$ based on this heurisitc, the remaining concepts in $\mathcal{H}$ are assigned to $\mathcal{H}_{\text{train}}$.

- Complexity: This split partitions into train and test based on length of the postfix serialization of the concept. Specifically, concepts shorter than 10 tokens are placed in $\mathcal{H}_{\text{train}}$ and longer concepts are placed in $\mathcal{H}_{\text{test}}$.

## D    CREATING SUPPORT AND QUERY SETS

We next explain how we go from the initial dataset $\mathcal{U}$ – which contains a large number of images, schema and sounds – and a concept space $\mathcal{H}_{\text{train}}$ and $\mathcal{H}_{\text{test}}$, to a dataset for meta learning. To create the training/validation/test sets for models, we sample a series of episodes, each containing a support set and a query set. We illustrate the sampling procedure for a training episode below:

**Support Set Sampling with Hard Negatives**

1. Pick a concept $h \sim p_{\text{train}}(h)$, with a preference for shorter hypotheses being more frequent based on the weights used to define the prior Appendix B.3

2. Pick 5 images ($P$), uniformly at random from $\mathcal{U}$ such that $h(\mathbf{u}^s) = 1$, where the concept is evaluated on the schema to determine the label Appendix B.4

3. Identify other concepts $h' \in \mathcal{H}$ s.t. $h(u(S)) = 1$ and $h' \neq h$

4. Pick images such that $h'(\mathbf{u}^s) = 1$ and $h(\mathbf{u}^s) = 0$ as negatives ($N$). If no such images exist, pick random images from $\mathcal{U}$ as negatives until we have 20 negatives.

5. Return $D_{\text{supp}} = P \cup N$.

The sampling procedure for the Query set iterates all the steps above (except step 1, where we choose the concept $h$). Step 3 and 4 outline a procedure for identifying hard negatives for training the model, by looking at other hypotheses which also explain a chosen set of positives $P$ and using them to clarify what the concept of interest is.

We give below an analogous procedure for easy negatives:

**Support Set Sampling with Easy Negatives**

1. Pick a concept $h \sim p_{\text{train}}(h)$, with a preference for shorter hypotheses being more frequent based on the weights used to define the prior Appendix B.3

2. Pick 5 images ($P$), uniformly at random from $\mathcal{U}$ such that $h(\mathbf{u}^s) = 1$, where the concept is evaluated on the schema to determine the label Appendix B.4

3. Pick 20 random images from $\mathcal{U}$ as negatives, N.

4. Return $D_{\text{supp}} = P \cup N$.

Similar to hard negatives, the sampling procedure for the Query set iterates all the steps above (except step 1, where we choose the concept $h$).

## E    REPRODUCIBILITY AND HYPERPARAMETERS

For all the models in Figure. 4 in the main paper, we use the following hyperparameters. All the modalities are processed into a set of objects $\{o_i\}_{i=1}^N$ where each $o_i \in \mathbb{R}^{64}$ for image and sound

models while for schema $o_i \in \mathbb{R}^{96}$. Further, the we use a learning rate of `1e-4` for image models, `1e-3` for schema models, and `5e-5` using the best learning rate for each modality across an initial sweep. The batch size for image and sound models is 8 episodes per batch, while for schema we use a batch size of 64. All models use the Adam optimizer. The overall scene representation across all the modalities is set to 256, that is, $\mathbf{u} \in \mathbb{R}^{256}$. All our models are initialized with the method proposed in (Glorot & Bengio, 2010), which we found to be crucial for training relation networks well. The initial representation from the first stage of the encoder (Fig. 4 in main paper) with the objects for images has a size of `10x8` *i.e.* there are 80 objects, while for sound representations have `38` objects. In the schema case the number of objects is the ground truth number of objects which is provided as input to the model.

We trained all the models on the training set, and picked the best performing checkpoints on training – measured in terms of mAP– to report all the results in the main paper. Our image and schema models are trained for 1 million steps (16 epochs for images, 128 epochs for schemas) while the sound models are trained for 500K steps, and checkpoints are stored after every 30K steps.

All our models fit on a GPU with 16GB capacity except the relation network trained with image inputs, which needs at 32GB GPU. We use the pytorch framework to implement all our models.

## F    MODEL ARCHITECTURES FOR POOLING

In this section we detail the exact architectures used for the different pooling operations we consider in this paper as shown in Figure 4 center panel. We first establish some notation. Let $o_i \in \mathbb{R}^K$ be the output object feature from the modality specific encoder (Figure 4, left panel), and let us denote by $O = \{o_i\}_{i=1}^{|O|}$ the set of features for each of the objects in the scene, which includes optional position information indicating where the object is present in the scene (Figure 4). Let $N$ be the requested dimensionality of the feature space from the pooling operation. Given this, we can describe the pooling operations used as follows:

- **avg-pool**: We first average the representations across all the objects $\{o_i\}_{i=1}^{|O|}$ and then pass the averaged representation through an MLP with `256 x 512 x 384 x N` units with batch normalization and rectified linear unit nonlinearity in the hidden layers.

- **concat**: We first concatenate all the object representations in $O$, followed by an MLP with `256 x 512 x 256 x N` units with batch normalization and rectified linear units nonlinearity in the hidden layers.

- **relation-net**: For relation networks, following (Santoro et al., 2017) we use relative position encoding that captures the relative positioning of the objects in a scene for image and sound modalities, and use the location information already present in the schema modality. Based on this, in the terminology of Santoro et al. (2017) our $g()$ MLP has `256 x 256 x 256 x 256` hidden units with rectified linear unit non linearity and batch normalization whereas our $f()$ MLP has `256 x 256 x N` units with recitifed linear unit non linearlity and batch normalization in the middle (non-output) layers. Different from the original paper, we do not use dropout as we did not observe any overfitting in our experiments.

- **transformer**: We use a 2-head multi-head attention layer stacked 4 times, with the feedforward network dimenstions set to 512. After forwarding through this module, we take the output vectors $o_i'$ for each object processed through these initial layers and pool across objects by doing `max(), mean(), sum(), min()` operations and concatenating their outputs, similar to previous work by Wang et al. (2019). The final representation then does a linear projection of this concatenated vector to $N$, the dimensionality expected from the pooling module.

## G    ADDITIONAL RESULTS

### G.1    HYPERPARAMETER SWEEPS – OBJECT FEATURE DIMENSIONS

We next show the hyperparameter sweeps for image models in determining the choice of the dimensionality to represent each object $o_i$ for our image models (Figure 14). The same choice of

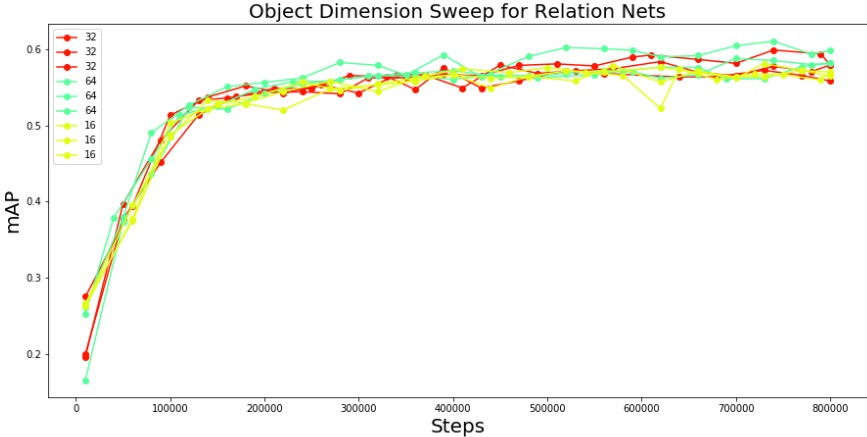

Figure 14: mAP on validation for hard negatives (y-axis) vs number of training steps (x-axis) for relation network models on images with different dimensionality for the object embedding $o_i$.

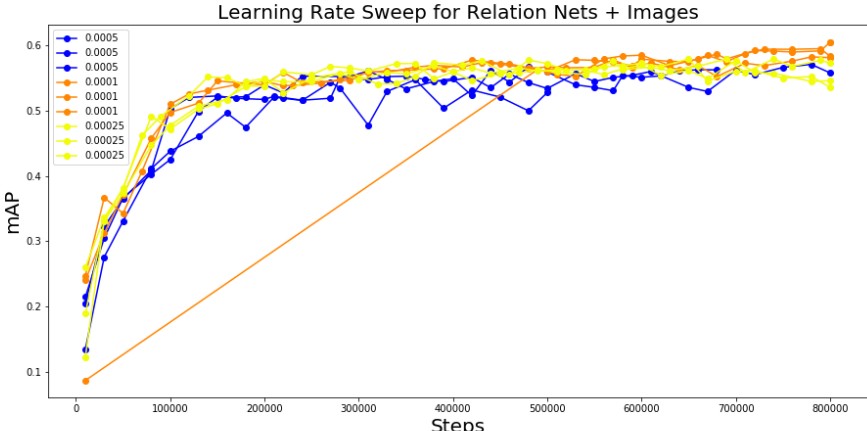

Figure 15: mAP on validation for hard negatives (y-axis) vs number of training steps (x-axis) for relation network models on images with different learning rates.

dimensionality was replicated for sound models. In our initial sweeps, on the Concept IID split, across different choices of the dimensionality of objects, we found relation networks to outperform concat and global average pooling models substantially, and thus we picked the object dimensions based on what performs best for relation networks since overall we are interested in the best possible choice of models for a given split and modality. Based on the results in Figure 14 we picked $o_i \in \mathbb{R}^{64}$.

### G.2 IMAGE RELATION NETWORKS LEARNING RATE SWEEPS

We picked the learning rate for image models based on the best performing image relation network model, which across an initial sweep we found to yeild the best class of models. Figure 15 shows the performance of the models across learning rates of `{1e-4, 5e-4, 2.5e-4}`.

### G.3 SWEEP ON USE OF LANGUAGE

As explained in the main paper (Figure. 4), the parameter $\alpha$ controls the tradeoff between the query accuracy and the likelihood of the concept expressed as a prefix string. We generally found across a broad range of values in `{0.0, 0.01, 0.10, 1.0}` the models generally performed the best at $\alpha = 1.0$. Our initial experiments with $\alpha = 10.0$ suggested substantially worse performance so we discarded it from the sweep. See Figures 16 to 18 for the corresponding results.

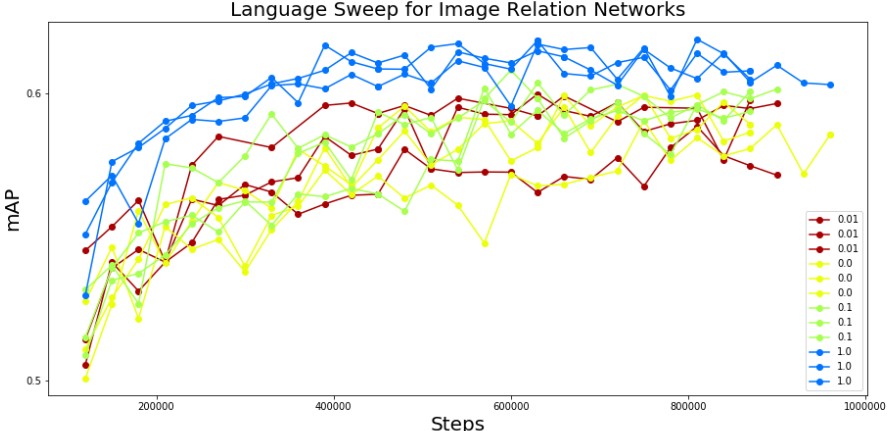

Figure 16: mAP on validation for hard negatives (y-axis) vs number of training steps (x-axis) for relation network models on images with different amounts of language usage by varying the parameter $\alpha$.

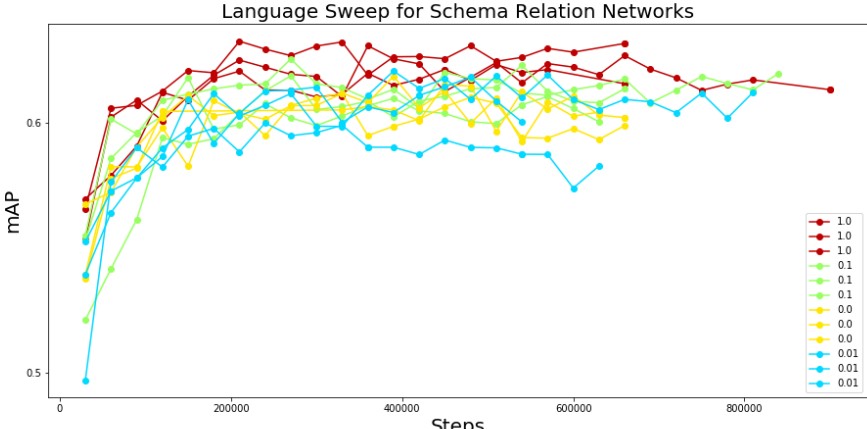

Figure 17: mAP on validation for hard negatives (y-axis) vs number of training steps (x-axis) for relation network models on schemas with different amounts of language usage by varying the parameter $\alpha$.

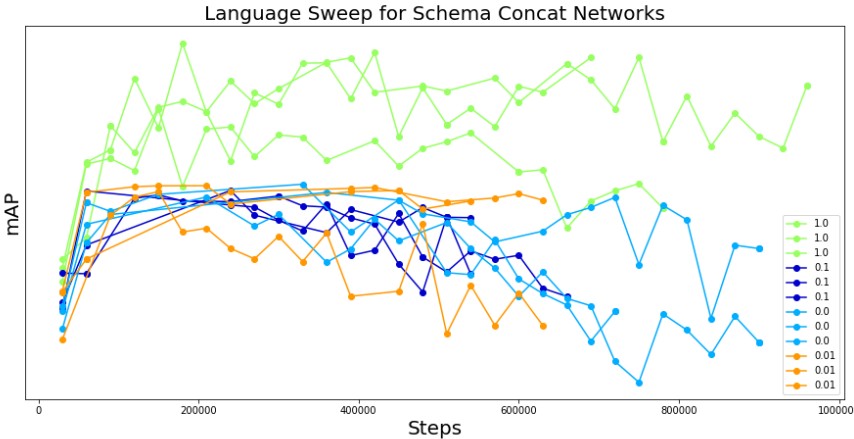

Figure 18: mAP on validation for hard negatives (y-axis) vs number of training steps (x-axis) for concat pooling models on schemas with different amounts of language usage by varying the parameter $\alpha$.

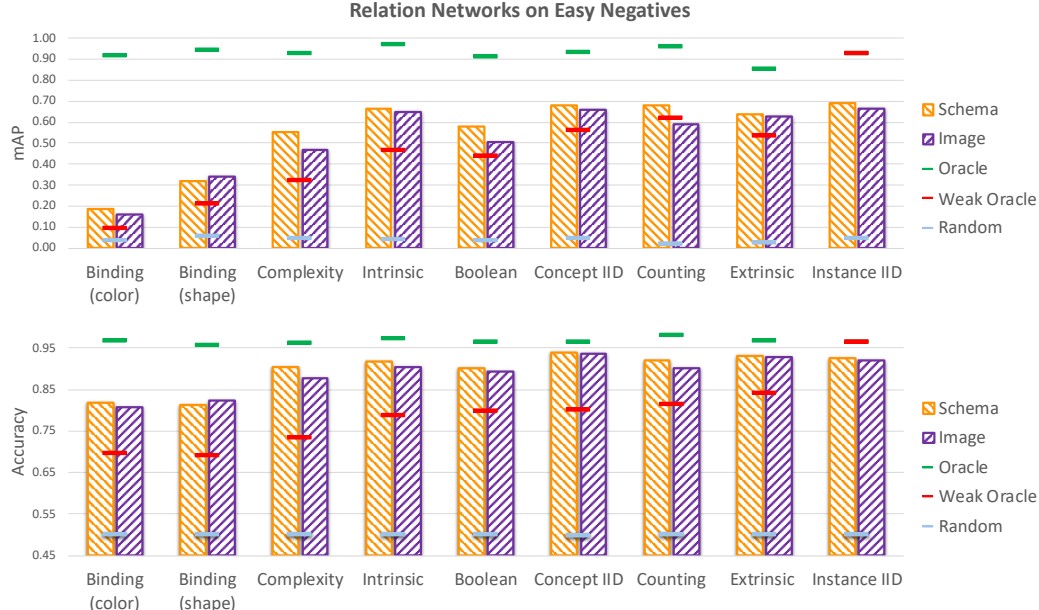

Figure 19: mAP (top) and accuracy (bottom) metrics for the different splits presented in the paper when easy negatives are used (ordered by their corresponding `comp gap`). **Yellow:** shows the schema relation-net model while **purple:** shows the image relation-net model. Notice that compared to Fig. 5 in the main paper, the `comp gap` is smaller and the models appear to be substantially closer to the strong oracle in this setting compared to when we use hard negatives.

## G.4 RESULTS ON EASY NEGATIVES

In Figure 19 we show results for the relation-net model on various splits, where easy negatives are used to populate the support and query sets during training and evaluation, unlike the case of hard negatives discussed in the main paper (Figure. 5). Notice that the compositionality gap (`comp gap`) is lower in general for easy negatives compared to the hard negatives as reported in the main paper. Further, we find that the best models are substantially closer to the strong oracle compared to Figure. 5 main paper, showing that on the easier, less compositional task it is easier for machine learning models to approach the strong oracle (especially in terms of accuracy). Finally, it is interesting to note that with easy negatives it appears that the best models outperform the weak oracle on the Counting split, while with the hard negatives one finds that the models are worse than the weak oracle, suggesting poor generalization for counting.

## G.5 FINER $\alpha$ SWEEP FOR COUNTING

Finally, we ran a finer alpha sweep for the Counting split since it appeared on our initial sweep that the counting split was not performing better with language. Concretely, we ran a new set of experiments sweeping over $\alpha$ values of {0.01, 0.10, 1.0, 5.0, 10.0, 100.0}. Across this broader range of values, we found models still did not show any statistically significant gains from using language *v.s.* not for the Counting split.

## G.6 CHOICE OF METRIC: MAP *v.s.* ACCURACY

In general, the mAP metric opens up a larger `comp gap` for the various splits than indicated by CBA. For example, with hard negatives, while CBA indicates a gap of 14.2% for Counting compared to 0% for Instance IID, mAP suggests a gap of 34.4% for Counting relative to 0% for Instance IID. For the Binding (color) split its 86.5% `comp gap` (mAP) *v.s.* 34.0% for CBA. mAP, while being more expensive to compute evaluates more thoroughly to test if a concept $h$ is truly learnt by the model, by probing its performance on a large, representative set of negatives $\mathcal{T}$, providing a more stringent test of compositional generalization.

Table 1: Performance on meta-test, sorted based on mAP (in %) on Binding (color) with hard negatives

| modality | pooling | mAP (mean) | mAP (std) |
|---|---|---|---|
| schema | transformer | 15.9 | 0.9 |
| schema | relation-net | 15.3 | 0.9 |
| image | avg-pool | 15.1 | 0.4 |
| image | relation-net | 14.8 | 0.7 |
| schema | avg-pool | 14.4 | 1.2 |
| image | transformer | 14.2 | 0.2 |
| schema | concat | 14.0 | 0.7 |
| image | concat | 13.5 | 0.4 |
| sound | avg-pool | 9.4 | 0.6 |
| sound | relation-net | 9.2 | 0.3 |
| sound | concat | 8.0 | 0.6 |

Table 2: Performance on meta-test, sorted based on mAP (in %) on Boolean with hard negatives

| modality | pooling | mAP (mean) | mAP (std) |
|---|---|---|---|
| schema | relation-net | 51.1 | 1.4 |
| schema | avg-pool | 48.0 | 2.1 |
| schema | concat | 47.1 | 2.1 |
| schema | transformer | 46.9 | 1.1 |
| image | transformer | 44.0 | 2.2 |
| image | relation-net | 38.4 | 1.0 |
| image | avg-pool | 36.3 | 2.3 |
| image | concat | 29.6 | 2.4 |
| sound | relation-net | 25.5 | 0.8 |
| sound | avg-pool | 24.1 | 1.6 |
| sound | concat | 22.3 | 1.1 |

## G.7 DETAILED RESULTS ON ALL THE SPLITS IN HARD NEGATIVES SETTING

In this section we provide the full results of all of the tested models on each of the splits considered in the paper, in the hard negatives setting. Tables 1-9 show the results of different models (sorted in a descending order based on mAP for each of the splits considered in the paper, in the case where models do not have access to language.

## G.8 DETAILED RESULTS ON ALL THE SPLITS IN EASY NEGATIVES SETTING.

In this section we provide the full results of all of the tested models on each of the splits considered in the paper, in the easy negatives setting. Tables 10-18 show the results of different models (sorted in a descending order based on mAP for each of the splits considered in the paper, in the case where models do not have access to language. Note that we did not evaluate transformer models or sound models in this setting as this is qualitatively less interesting than the hard negatives setting and is not the main focus of the paper.

Table 3: Performance on meta-test, sorted based on mAP (in %) on Intrinsic with hard negatives

| modality | pooling | mAP (mean) | mAP (std) |
|---|---|---|---|
| schema | transformer | 57.9 | 0.6 |
| schema | relation-net | 55.1 | 0.9 |
| image | transformer | 54.1 | 3.0 |
| schema | avg-pool | 53.3 | 3.0 |
| schema | concat | 52.2 | 3.5 |
| image | relation-net | 47.6 | 7.5 |
| image | avg-pool | 34.5 | 0.4 |
| image | concat | 34.0 | 1.0 |
| sound | concat | 29.4 | 1.6 |
| sound | avg-pool | 26.5 | 2.3 |
| sound | relation-net | 24.3 | 2.4 |

Table 4: Performance on meta-test, sorted based on mAP (in %) on Concept IID with hard negatives

| modality | pooling | mAP (mean) | mAP (std) |
|---|---|---|---|
| image | transformer | 60.8 | 0.4 |
| schema | relation-net | 60.7 | 0.3 |
| schema | transformer | 59.8 | 0.2 |
| image | relation-net | 58.5 | 0.7 |
| image | avg-pool | 56.7 | 1.0 |
| schema | avg-pool | 54.6 | 1.3 |
| schema | concat | 53.3 | 0.9 |
| image | concat | 53.0 | 0.6 |
| sound | concat | 23.6 | 2.7 |
| sound | avg-pool | 22.7 | 2.3 |
| sound | relation-net | 21.8 | NaN |

Table 5: Performance on meta-test, sorted based on mAP (in %) on Instance IID with hard negatives

| modality | pooling | mAP (mean) | mAP (std) |
|---|---|---|---|
| schema | relation-net | 63.9 | 0.4 |
| image | transformer | 61.5 | 0.9 |
| schema | transformer | 59.8 | 0.5 |
| image | relation-net | 58.6 | 1.3 |
| image | avg-pool | 58.1 | 0.7 |
| schema | avg-pool | 57.4 | 0.9 |
| image | concat | 57.1 | 0.9 |
| schema | concat | 57.1 | 1.4 |
| sound | avg-pool | 23.5 | 1.4 |
| sound | relation-net | 22.3 | 2.3 |
| sound | concat | 21.7 | 0.5 |

Table 6: Performance on meta-test, sorted based on mAP (in %) on Extrinsic with hard negatives

| modality | pooling | mAP (mean) | mAP (std) |
|---|---|---|---|
| image | transformer | 62.1 | 0.8 |
| schema | transformer | 60.9 | 0.6 |
| image | relation-net | 60.8 | 1.1 |
| image | concat | 60.8 | 0.7 |
| schema | relation-net | 59.8 | 0.4 |
| image | avg-pool | 59.4 | 1.3 |
| schema | concat | 54.3 | 2.3 |
| schema | avg-pool | 53.4 | 1.5 |
| sound | avg-pool | 25.7 | NaN |
| sound | concat | 23.2 | 1.4 |
| sound | relation-net | 18.3 | 1.7 |

Table 7: Performance on meta-test, sorted based on mAP (in %) on Complexity with hard negatives

| modality | pooling | mAP (mean) | mAP (std) |
|---|---|---|---|
| schema | relation-net | 52.8 | 0.7 |
| schema | transformer | 49.7 | 0.6 |
| schema | avg-pool | 48.4 | 1.0 |
| schema | concat | 48.0 | 1.2 |
| image | avg-pool | 45.8 | 0.4 |
| image | transformer | 45.6 | 0.2 |
| image | relation-net | 45.4 | 0.4 |
| image | concat | 42.6 | 3.4 |
| sound | concat | 26.0 | NaN |
| sound | relation-net | 23.3 | 1.0 |
| sound | avg-pool | 23.0 | 1.7 |

Table 8: Performance on meta-test, sorted based on mAP (in %) on Binding (shape) with hard negatives

| modality | pooling | mAP (mean) | mAP (std) |
|---|---|---|---|
| schema | transformer | 30.9 | 0.6 |
| schema | relation-net | 30.4 | 1.5 |
| image | relation-net | 29.1 | 1.8 |
| image | avg-pool | 28.7 | 1.0 |
| image | transformer | 28.7 | 0.1 |
| schema | avg-pool | 28.3 | 1.9 |
| image | concat | 27.0 | 0.8 |
| schema | concat | 27.0 | 0.8 |
| sound | relation-net | 17.5 | 0.4 |
| sound | concat | 17.4 | 1.3 |
| sound | avg-pool | 16.2 | 1.6 |

Table 9: Performance on meta-test, sorted based on mAP (in %) on Counting with hard negatives

| modality | pooling | mAP (mean) | mAP (std) |
|---|---|---|---|
| schema | relation-net | 57.4 | 4.8 |
| schema | avg-pool | 55.0 | 6.4 |
| schema | concat | 50.0 | 3.2 |
| image | avg-pool | 48.5 | 1.3 |
| schema | transformer | 40.7 | 0.4 |
| image | relation-net | 40.3 | 2.0 |
| image | concat | 38.5 | 2.8 |
| image | transformer | 32.4 | 1.4 |
| sound | concat | 13.8 | 2.4 |
| sound | avg-pool | 13.4 | 1.0 |
| sound | relation-net | 13.0 | 1.7 |

Table 10: Performance on meta-test, sorted based on mAP (in %) on Binding (color) with easy negatives

| modality | pooling | mAP (mean) | mAP (std) |
|---|---|---|---|
| schema | relation-net | 18.6 | 1.6 |
| image | relation-net | 16.3 | 0.8 |
| image | avg-pool | 16.3 | 0.5 |
| image | concat | 15.6 | 0.5 |
| schema | avg-pool | 15.5 | 0.5 |
| schema | concat | 15.5 | 0.5 |

Table 11: Performance on meta-test, sorted based on mAP (in %) on Boolean with easy negatives

| modality | pooling | mAP (mean) | mAP (std) |
|---|---|---|---|
| schema | relation-net | 58.1 | 2.0 |
| schema | avg-pool | 55.0 | 2.1 |
| schema | concat | 54.7 | 2.2 |
| image | relation-net | 50.7 | 7.3 |
| image | avg-pool | 42.9 | 1.6 |
| image | concat | 40.9 | 1.7 |

Table 12: Performance on meta-test, sorted based on mAP (in %) on Counting with easy negatives

| modality | pooling | mAP (mean) | mAP (std) |
|---|---|---|---|
| schema | relation-net | 67.9 | 8.5 |
| schema | avg-pool | 64.4 | 7.3 |
| image | avg-pool | 62.2 | 6.6 |
| image | concat | 61.2 | 6.9 |
| schema | concat | 60.8 | 6.7 |
| image | relation-net | 58.8 | 5.7 |

Table 13: Performance on meta-test, sorted based on mAP (in %) on Extrinsic with easy negatives

| modality | pooling | mAP (mean) | mAP (std) |
|---|---|---|---|
| schema | relation-net | 63.7 | 1.4 |
| image | relation-net | 62.9 | 1.1 |
| image | concat | 61.8 | 2.4 |
| image | avg-pool | 61.5 | 1.4 |
| schema | avg-pool | 57.5 | 1.2 |
| schema | concat | 57.2 | 1.5 |

Table 14: Performance on meta-test, sorted based on mAP (in %) on Intrinsic with easy negatives

| modality | pooling | mAP (mean) | mAP (std) |
|---|---|---|---|
| schema | relation-net | 66.5 | 4.8 |
| image | relation-net | 64.9 | 5.3 |
| image | avg-pool | 64.4 | 4.5 |
| schema | avg-pool | 63.1 | 4.7 |
| image | concat | 62.7 | 4.5 |
| schema | concat | 60.8 | 4.6 |

Table 15: Performance on meta-test, sorted based on mAP (in %) on Concept IID with easy negatives

| modality | pooling | mAP (mean) | mAP (std) |
|---|---|---|---|
| schema | relation-net | 68.2 | 1.0 |
| image | relation-net | 65.8 | 0.4 |
| image | avg-pool | 65.1 | 0.4 |
| image | concat | 64.7 | 0.4 |
| schema | avg-pool | 63.3 | 0.4 |
| schema | concat | 61.9 | 0.2 |

Table 16: Performance on meta-test, sorted based on mAP (in %) on Instance IID with easy negatives

| modality | pooling | mAP (mean) | mAP (std) |
|---|---|---|---|
| schema | relation-net | 69.0 | 3.6 |
| image | avg-pool | 66.8 | 3.3 |
| image | relation-net | 66.5 | 3.2 |
| image | concat | 65.4 | 3.4 |
| schema | avg-pool | 63.5 | 3.1 |
| schema | concat | 63.3 | 3.1 |

Table 17: Performance on meta-test, sorted based on mAP (in %) on Complexity with easy negatives

| modality | pooling | mAP (mean) | mAP (std) |
|---|---|---|---|
| schema | relation-net | 55.1 | 1.8 |
| schema | avg-pool | 51.9 | 2.0 |
| schema | concat | 51.3 | 1.9 |
| image | avg-pool | 49.0 | 1.6 |
| image | concat | 48.2 | 1.7 |
| image | relation-net | 46.6 | 1.5 |

Table 18: Performance on meta-test, sorted based on mAP (in %) on Binding (shape) with easy negatives

| modality | pooling | mAP (mean) | mAP (std) |
|---|---|---|---|
| image | relation-net | 33.9 | 2.4 |
| schema | avg-pool | 32.1 | 1.7 |
| image | avg-pool | 31.9 | 1.8 |
| schema | relation-net | 31.7 | 1.6 |
| image | concat | 31.3 | 1.7 |
| schema | concat | 31.0 | 1.5 |

