# OpenReview forum: "CURI: A Benchmark for Productive Concept Learning Under Uncertainty"
_ICLR.cc/2021/Conference — Reject_

### Official Review · AnonReviewer2 · 2020-10-27
**Interesting Question with Design Flaws and Missing References**

**Rating:** 5
**Confidence:** 4

**Review:**

This work proposes the CURI dataset to measure productive concept learning under uncertainty. The dataset is designed using a concept space defined by a language and formulated as a few-shot meta-learning problem to tell apart in-concept samples from out-of-concept samples. The authors also design several out-of-generalization data splits that test models' ood generalization performance. Together with an oracle model, the authors show using the prototypical network that the compositional concept learning and reasoning problem in CURI is challenging.

---

1. The proposed dataset touches the problem of generalizable concept learning, which is important, as more and more methods have been shown to be able to well fit the iid distributions.
2. The dataset is well-designed in terms of generalization. It explicitly separates the concept space and proposes several aspects for the generalization problem.
3. A good metric for the generalization gap that can measure how difficult each data split is by using oracle models.

---

1. One of the major problems I consider in this work is how to make sure the positive samples follow a unique concept. The authors mention that the actual concept space is unbounded and uncertain. Therefore, there could potentially be an infinite number of concepts where the positive samples are satisfied. Is it possible to make sure that the positive cases and the negative cases can prune all other concepts to make sure the actual concept tested in $D_{supp}$ is unique? If not, during testing, how to make sure a ''negative'' sample is truly negative rather than coincidentally satisfying the concept? If the problems can not be properly addressed, then the dataset may contain fundamental errors.
2. Another major problem is the missing references to existing few-shot meta-learning works. There has been a plethora of related papers and this work only compares with the most basic prototypical network. See the following references for examples. Besides, a couple of recent works also try to address the compositional learning problem. These models are also missing. Without sufficient experiments, it's hard to paint a full picture of the dataset.
3. The statements regarding CLEVR and PGM are inaccurate. The paper states that questions to ask need to be inferred in CURI, which I believe is not true because the question is simply asking a model to tell an odd one out. It's just that CLEVR provide questions that can be parsed to programs while CURI does not. For PGM, it's stated that "once the constraints of a puzzle are identified" ... However, the uncertainty problem and concept learning problem are right at identifying the constraints, same as in CURI where the concept description needs inferring.
4. Typos: Page 4 at the bottom "to to" -> "to", "yeild" -> "yield"
5. How is $P(H=h | D_{supp})$ defined? I think this could be a major part impacting the performance. If you have a proper reasoner to do it (say an oracle model), it doesn't make sense to have such a low accuracy. But if it is just a multi-way classification setup, then the performance can be poor.
6. A very important and highly related work is missing: Andreas, Jacob, Dan Klein, and Sergey Levine. "Learning with latent language." ACL 2018. This work is basically in the same setup as CURI and the models used, the methodology, the problem it considers "at least" need some in-depth discussions.

[1] Lake, Brenden M. "Compositional generalization through meta sequence-to-sequence learning." NeurIPS. 2019.
[2] Finn, Chelsea, Pieter Abbeel, and Sergey Levine. "Model-agnostic meta-learning for fast adaptation of deep networks." ICML. 2017.
[3] Bertinetto, Luca, et al. "Meta-learning with differentiable closed-form solvers." ICLR. 2018.
[4] Lee, Kwonjoon, et al. "Meta-learning with differentiable convex optimization." CVPR. 2019.
[5] Sung, Flood, et al. "Learning to compare: Relation network for few-shot learning." CVPR. 2018.

---

> ### Author Response · Authors · 2020-11-23
> **Initial responses to R2, Please also see "Responses to all reviewers"**
>
> We thank R2 for a detailed review and that they thought the dataset was “well designed” and the compositionality gap metric was “good” contribution. We clarify specific points below, and defer general issues which arose in the reviews to the “Response to all reviewers”.
>
> 1. “Unique concept, negatives might be coincidental making task ill defined”:
> While multiple concepts can and will be true for a support set, in our opinion it is a feature and not a bug, as one needs to reason which of those concepts might be more general and which ones might be specific (as illustrated by toy example in response to all reviewers). Given that the labeling is done using “one” of the possible concepts during testing, it is true that the negative might not in some sense be a true negative, that is, the task has a non-zero bayes error. However, this is not a fundamental issue -- even tasks such as image captioning or question answering can have ambiguity in the true output distribution thus have a non-zero bayes error, but are still legitimate tasks we are interested in as a community.  Infact, embracing such uncertainty in concept learning and providing a tractable evaluation setup with the oracles and compositionality gaps is one of our central contributions.
>
> 2. “Insufficient models”:
> * Reg. comparison to [1] we stress that our problem is a compositional, few-shot labelling task which uses meta-learning, while [1] is a compositional sequence prediction task which uses meta-learning. Given the tasks are different we dont see a direct comparison.
> *Reg. comparison to [4-5]: Ours is a compositional reasoning benchmark which frames the problem in the language of meta-learning, and not a meta-learning oriented benchmark such as [A]. This is not merely an academic difference, [A] has a completely different experimental setup, protocol and evaluation setting spanning different number of shots, image datasets etc. and thus requires a comparison of different *meta-learning* approaches. In contrast, since ours is a reasoning benchmark situated in the meta-learning context, we pick a state of the art meta-learner and study the impact of modeling choices more relevant to *compositional reasoning* than *meta-learning*. In this context we already compare to a number of models spanning relation networks, global average pooling, transformers, concatenation, different modalities, usage of language vs. not.
>
> 3. “Statements regarding CLEVR and PGM are not true”
> Please see our response to all reviewers for a clarification on the task setup with a toy example. The input to the model is simply Dsupp, which is a support set with positive and negative images. If the question was given, one would indeed have to tell apart the odd one out, but the question is not actually given, which is where the “reasoning under uncertainty” aspect becomes important.
> Reg. PGM: We are not sure we understand what the reviewer means by “uncertainty problem and concept learning problem are right at identifying the constraints”. Would be great if they could clarify. We reiterate that PGM is a deductive reasoning task one does not need to reason between competing concepts with a view towards generalization (as explained in toy example).
>
> 4. “How is P(H=h|Dsupp) defined? More complex oracle model might do better”
> Page 6 last paragraph describes how P(H=h| Dsupp) is defined. It is essentially an LSTM sequence model which maximizes the likelihood of the prefix serialization of the concept given the prototypes for the positive and negative examples in Dsupp. It would indeed be an interesting direction for future work on the dataset to utilize a more complex reasoning process in defining the likelihood term, but is out of scope for current paper.
>
> 5. Comparison to [B]: Although Andreas et.al. consider a very related learning setup, their major goal is to treat language as the space over which classification is done and demonstrate its use as a latent variable. To this end, they induce novel classifiers from examples via the space of linguistic expressions. Our goals differ in evaluating systematic generalization in productive concept learning in models that do not have access to the latent language, although we compare with language-augmented models as well. Concretely, this difference manifests in the various splits and the associated compositionality gaps, strong and weak oracles, and an explicit problem setup of studying reasoning under uncertainty which are not considered by [B]. We will add this to the paper.
>
> References:
> [A]: Triantafillou, Eleni, Tyler Zhu, Vincent Dumoulin, Pascal Lamblin, Utku Evci, Kelvin Xu, Ross Goroshin, et al. 2019. “Meta-Dataset: A Dataset of Datasets for Learning to Learn from Few Examples.” arXiv [cs.LG]. arXiv. http://arxiv.org/abs/1903.03096.
> [B]: Andreas, Jacob, Dan Klein, and Sergey Levine. 2017. “Learning with Latent Language.” arXiv [cs.CL]. arXiv. http://arxiv.org/abs/1711.00482.

---

### Official Review · AnonReviewer1 · 2020-10-28
**Initial Review**

**Rating:** 6
**Confidence:** 3

**Review:**

Summary:

The following work presents a CLEVR-based compositionality benchmark. The task of the model is to verify logical statements about an image, and in order to achieve such, must learn how to map individual statements to a composition of functions over the image checking for color, placement, shape, etc. Specific to this dataset is that it is explicitly few-shot, which forces the models to generalize very quickly and to infer under uncertainty.


Strengths:

-Introduction of novel compositionality gap to measure the extent to which any learned solution needs to extrapolate on each train/test split

-Dataset is very carefully split in multiple different ways to individually test for generalization across various aspects

-Extensive analysis, testing a wide range of the standard baseline methods

Weaknesses and Concerns:

-Like CLEVR, the statements are generated based on a grammar over a controlled set of relations/attributes/terms. While this is necessary for ruling out linguistic complexities from the analysis, one wonders how useful the dataset will be in the long run. Specifically because the dataset is setup to be used in a few-shot setting, models that perform increasingly better will do so by incorporating strong priors that match the true distribution of the dataset, at which point would we not simply be overfitting to this dataset due to its limited linguistic scope as we did for CLEVR?

-It's not clear to me what the  benefit of the audio modality is. Specifically, I imagine that there are a number of different ways to encode the relational information in audio, with each possibly giving different results. Any conclusions are likely specific to the particular encoding rather than the audio modality as a whole.

Additional questions:

-What were the main conclusions that we can draw from this dataset that would have been impossible in CLEVR?

Typos:

Appendix Figure 18: y axis has no tic marks

Overall, I like the clean setup of the dataset's split and the some of the few-shot analysis on exsiting models. However, the novelty of the dataset as a whole is limited, as it is effectively a modification of CLEVR. It is not yet clear to me that the analysis has yielded any particularly groundbreaking insights, though I would love for the authors to correct me on that. Most importantly, I have strong doubts about the long-term usefulness of this dataset as a benchmark before we start overfitting, given its limited scope and few-shot setting.

---------------------------------------------------------------------------------------------

Post rebuttal update:

After reading the clarifications from the authors, it is now more clear that the dataset is about the learnability of certain hypotheses as opposed to that of test-time verification. I am generally satisfied with the responses given by the authors, and willing to buy the statement that having a per-concept breakdown of learnability can be seen as a feature rather than a weakness pertaining to the use of a "toy" dataset. I think there are some valid insights as provided by this work, though I am more skeptical regarding the superiority of transformers on disentanglement tasks as the improvement gap was relatively small and there are various implementation details in the used transformers and relationship networks that could presumably shrink or even invert that gap.

Overall, I think the writing clarity is still a significant concern. Several detailed passes over the paper were necessary just to get the general picture of what was going on. I think the toy example provided in the author's response is certainly helpful, and should improve the paper in this area somewhat.

Based on the above, I am upgrading my rating to marginally above threshold.

---

> ### Author Response · Authors · 2020-11-23
> **Initial Responses to R1, please also see "Response to all Reviewers"**
>
> We thank the reviewer for raising thoughtful points around the utility of the benchmark over CLEVR. We have included a more general response related to the task and importance of the task and how it is different from CLEVR in the response to all reviewers. Here we address more specific issues:
>
> -- The reviewer says “The task of the model is to verify logical statements about an image, and in order to achieve such, must learn how to map individual statements to a composition of functions over the image checking for color, placement, shape, etc.”
>
> This does not appear to be a factually true statement about the dataset, in the sense that the statement or logical expression to be verified is not actually provided as input to the model, and thus the task of the model is not to “verify logical statements”. The model must reason over the (potentially combinatorial) space of likely explanations for the images it receives and must behave as if marginalizing over the possibilities and making predictions. Please see the general response for more clarification about the task with a toy example.
>
> -- “Overfitting to limited linguistic scope like CLEVR”
>
> As we understand it, this perhaps stems from the confusion about the task specification (above). There is no language input (by default, except in one set of experimental settings) given in general to the model. We are testing if a model is able to behave as if it understands which all concepts (from a large space of concepts) apply to a given set of images. In order to do this, we need to situate ourselves in a setting where we know which concepts apply to a given situation in order to evaluate such a behavior in models. Thus, we do not expect that there will be overfitting to the language. In fact, understanding the priors of the world and reasoning about which among a combinatorially large space of concepts applies in a given situation is something we actively want to test, and should be considered a feature, not a bug in our opinion.
>
> -- “Benefit of audio modality”
>
> Audio modality has superimposition in the signals which is not present in the image and schema modalities, providing a different challenge for representation learning. Fig. 4 (model schematic) shows that for each modality we learn an encoder and a relational reasoning module on top of the encoding. Thus we can say something more general about representation learning based on the trends for a given modality over a range of encodings we work with (4 in our case). Such trends might not be too specific to the choice of the relational reasoner, but might indicate something more generic about the modality itself (marginalizing over choices of relational reasoning).
>
> -- “What were the main conclusions that we can draw from this dataset that would have been impossible in CLEVR?”
>
> We emphasize that the task itself is very different, and is not simply a verification task where the statement is given and one has to verify it on an image, which indeed would be a very similar task to CLEVR and might be where the reviewer’s concern stems from. The proposed task requires a notion of compositional reasoning under uncertainty (see response to all reviewers) which is not present in CLEVR, and thus it is difficult to make an apples to apples comparison. This is further evidenced by the fact that state of the art models such as relation networks which perform really well on the CLEVR dataset do not perform as well on the CURI dataset, indicating that the nature of reasoning required appears to be substantially different.
>
> -- “No groundbreaking insights”
>
> We feel in general it is a high bar for a dataset or benchmark paper to also include groundbreaking insights looking at previous benchmark papers in the community such as ImageNet, VQA, CLEVR, etc. which proposed interesting tasks for the community to tackle, and the original papers did not necessarily also supply “groundbreaking” insights. However, we do think there are some interesting insights from the results in the paper, such as:
> *This is the first test of variable binding we are aware of (as cited in Page. 1) in the machine learning literature and it appears to be a
>    really challenging split for models (Fig. 5)
> *Transformers seem to do really well at disentangling, which could explain their strong performance for word models/ NLP
> *Transformers seem to really do badly at counting, which is an interesting finding to investigate further in other scenarios

---

### Official Review · AnonReviewer4 · 2020-10-29
**compositional reasoning under uncertainty**

**Rating:** 6
**Confidence:** 3

**Review:**

This work introduces the "compositional reasoning under uncertainty" benchmark.

This is motivated by the observations that standard [classification] benchmarks:
 1) consider only a fixed set of category labels,
 2) do not evaluate compositional concept learning
 3) do not explicitly capture a notion of reasoning under uncertainty

The work proposes a task that requires several aspects of compositional reasoning, such as boolean operations, counting, etc. The author(s) introduce a notion of a "compositionality gap" to quantify the difficultly of each generalization type.

I buy the vision of the work and I consider it a promising direction. However, I am not completely convinced by the work and its contribution to the existing works, especially the recent related work.

What is the source of "uncertainty" in this challenge? What kind of "uncertainty" are we dealing with? The only place that you refer to it is the 2nd paragraph of your intro, but otherwise, I don't see anything else (while there are 18 mentions of "uncertainty" in section 1-2, there is only one mention of "uncertainty" in the whole main body section 3-5 and that is only in the title of Section 3!)
Why I am not convinced this is a good (useful?) notion of "reasoning under uncertainty". Just because two formulas/concepts can describe one scene, implies "uncertainty"? If so, what is the "reasoning" under this defined "uncertainty"?

In terms of writing: the intro/abstract are very clearly written and are easy to follow. But the rest of the work could be improved.

Overall, good work but requires a bit of work polishing to make the contributions stronger.

==============

Minor issues: "Producitivity": typo?

I felt that you're citing so many works that at times make it difficult to read.

"disentangling": clarify or cite. It's not clear how it's related to the tasks listed in Fig 5.

Define "productive generalization" and "systemic generalization".

=======
UPDATE after reading author response:  The authors' example answered my question about the nature of the "uncertainty" in their setting. I have increased my confidence score and have retained my marginally-positive evaluation.

---

> ### Author Response · Authors · 2020-11-23
> **Responses to R4, please also see "Response to all Reviewers"**
>
> We thank you for the thoughtful review, and are encouraged that you buy the vision of the work. Please see “Response to all reviewers” for a detailed description providing intuitions of the kind of reasoning under uncertainty captured by the benchmark. We are updating the paper to reflect this intuition with real examples from the dataset as well, thanks for bringing up this issue! Also, we will fix the other sections of the paper (outside of intro/abstract) to make them read more smoothly, and are happy to include more specific writing related feedback on those sections.
>
> Minor Points
> ==========
> We wanted to be on the more complete side when citing related work, but will revise the paper to ensure it does not get in the way of the flow of the paper.
>
> Disentangling: We meant the ability to separate out properties of objects such as color or shape in independent dimensions in a latent representation, in the spirit of works such as [A]. We will add a citation to the paper.
>
> By productive/systematic generalization we mean the ability of models to generalize to novel compositions of atoms than those seen during training. For example, generalizing to "blue circles" having only seen "blue squares" and "green circles". This is related to the notion of systematicity [B]. We will add this citation to make it more clear.
>
> References:
> [A]: Higgins, Irina, Loic Matthey, Arka Pal, Christopher Burgess, Xavier Glorot, Matthew Botvinick, Shakir Mohamed, and Alexander Lerchner. 2016. “Beta-VAE: Learning Basic Visual Concepts with a Constrained Variational Framework.” Openreview.net › Forumopenreview.net › Forum. https://openreview.net/pdf?id=Sy2fzU9gl.
> [B]: Lake, Brenden M., and Marco Baroni. 2017. “Generalization without Systematicity: On the Compositional Skills of Sequence-to-Sequence Recurrent Networks.” arXiv [cs.CL]. arXiv. http://arxiv.org/abs/1711.00350.

---

### Author Response · Authors · 2020-11-23
**Response to all Reviewers**

We thank the reviewers for their time and detailed feedback, and are glad that the reviewers found our work to be in a “promising direction” (R4), has extensive analysis and baseline models (R1), and that the compositionality gap was a “good metric” (R1, R2). We include below a toy example which might be beneficial to the overall discussion about the benchmark and address more specific points in replies to each individual reviewer.

We hope the toy example below alleviates the concerns such as:

----
*“What is the notion of uncertainty in reasoning addressed by this task?” (R4)

*“What are the groundbreaking insights from this work that are not yielded by existing benchmarks like CLEVR?” (R1)

*“Statements about CLEVR and PGM are inaccurate” (R2)

----

**TOY EXAMPLE for CURI (R4, R1, R2)**

Consider the following positive examples for two related concepts:
* Concept 1: All characters are bold and *for all* “a’s” there is no “b” preceding it.
* Example Set 1: **aabb**, **ab**,**aabb**, **aaaabb**


* Concept 2: All characters are bold and *there exists* an “a” such that there is no “b” preceding it
* Example Set 2: **abaa**, **ab**, **aaba**, **aaaabb**

Consider being given the examples in Set 2 (**abaa**, **ab**, **aaba**, **aaaabb**) as the support set Dsupp. In such a case, as far as Dsupp is concerned, both 1 and 2 are equally valid concepts.  However, the key nuance is that they are very different in terms of generalization -- Concept 2 imposes less constraints on the space of strings (since it asks for a “there exists” as opposed to “for all”) and is more likely to be true, compared to Concept 1 which is more specific. We believe a compositional learner should understand such differences between concepts and capture them by placing appropriate probabilities on each possibility when making predictions, and it is precisely this notion of “reasoning under uncertainty” that the CURI benchmark captures.

We are adding a figure and text illustrating this property on an actual example from the dataset to the paper. Thanks for bringing this up!

**TOY EXAMPLE APPLIED TO CLEVR (R1,R2)**

In CLEVR, there is no such need to reason under uncertainty, since one is given an example (image/ string): **aabb** and a question, “Are all the characters bold and is every “a” such that there is no “b” preceding it?” and one has to answer “yes/no” (or multiple choice in general). Note that one did not have to consider an alternative concept -- with a view towards genralization -- when performing this reasoning process unlike what we had to do for CURI. Thus, CURI is a fundamentally different compositional learning task which requires reasoning under uncertainty.

**TOY EXAMPLE APPLIED TO PGM (R1,R2)**

PGM also does not require models to explicitly reason about the difference in generalization for Concept 1 and Concept 2 and to make predictions respecting such graded differences on held out data.

---

### Comment · Area_Chair1 · 2020-11-23
**Author Response**

Dear reviewers:

We are about to end the second discussion stage. Could you please check whether the authors have addressed your concerns and questions and potentially ask any further clarification questions?

Thank you, Your Area Chair

---

### Decision · Program_Chairs · 2021-01-07
**Final Decision**

**Decision:**

Reject

**Comment:**

This paper was reviewed by 3 experts in the field. The reviewers raised their concerns on lack of novelty, unconvincing experiment, and the presentation of this paper, While the paper clearly has merit, the decision is not to recommend acceptance. The authors are encouraged to consider the reviewers' comments when revising the paper for submission elsewhere.